# COT: CONSISTENT OPTIMAL TRANSPORT WITH APPLICATIONS TO VISUAL MATCHING AND TRAVELLING SALESMAN PROBLEMS

## ABSTRACT

This paper extends the traditional Optimal Transport (OT) to the Consistent Optimal Transport (COT), which accommodates more than two measures while maintaining transport consistency. The problem is formulated by minimizing the transport costs between pairs of measures and enforcing cycle-consistency among them. We introduce both the Monge and Kantorovich formulations of COT and derive an approximate solution through the addition of entropic and consistency regularization. An iterative projection algorithm, RCOT-Sinkhorn, is developed to enhance the Sinkhorn algorithm. In the visual multi-point matching task, our COT solver directly employs the cosine distance between learned point features from existing graph matching neural networks as the pairwise cost, achieving significant improvement in learning multiple matchings without further feature training. Additionally, based on COT, we present a new formulation for the Travelling Salesman Problem (TSP), termed TSP-COT. Regularization is used to relax the optimization, and the modified RCOT-Sinkhorn algorithm is applied to obtain the probability matrix of TSP routing. A post-process search method is then utilized to determine the TSP routes, and experiments validate the superiority of our approach. The code will be made available.

## 1 INTRODUCTION

Optimal transport (OT) (Peyre & Cuturi, 2019), as a fundamental mathematical tool, has been widely applied in numerous machine learning domains to learn the optimal transportation between source and target probability measures, including domain adaptation (Tzeng et al., 2017; Cui et al., 2018), generative models (Arjovsky et al., 2017), network design (Xu & Cheng, 2023), self-supervised contrastive learning (Caron et al., 2020; Shi et al., 2023), and long-tail recognition (Peng et al., 2021; Shi et al., 2024) etc. However, in many real-world scenarios, the traditional OT, which primarily focuses on transportation between two distributions, often falls short of handling complex situations involving multiple distributions like point matching (Wang et al., 2023). This limitation has spurred the need for a generalized form of OT to handle multiple distributions problems.

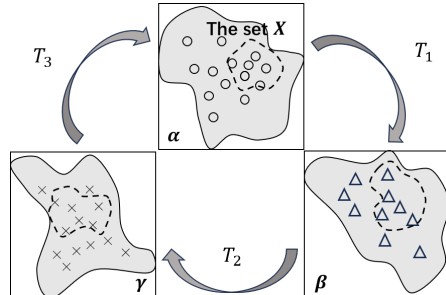

Figure 1: Consistent Optimal Transport (COT): three probability measures $\alpha, \beta, \gamma$, COT satisfies $T_{1\#}\alpha = \beta, T_{2\#}\beta = \gamma, T_{3\#}\gamma = \alpha$, and the cycle-consistency constraints $X = T_3(T_2(T_1(X)))$ given $X$ sampled from $\alpha$.

In this paper, we introduce the Consistent Optimal Transport (COT), which extends the capabilities of OT by accepting more than two measures as input with transport consistency. As illustrated in Fig. 1, the case considers three probability measures and it seeks to minimize the cost of three transportations while ensuring cycle-consistency among measures: specifically given a set $X$ sampled from the probability measure $\alpha$, the transportation mappings $T_1, T_2, T_3$ satisfy the condition $X = T_3 \circ T_2 \circ T_1(X)$. Following this setting, in this paper we propose Monge and Kantorovich formulations for COT, considering the challenges posed by cycle-consistency constraints in solving the problems. Specifically, we introduce the entropic

regularization transforming the hard cycle-consistency constraint into a regularizer in the objective function. The resulting regularized version of COT, can be approximately and efficiently solved using our proposed algorithms called RCOT-Sinkhorn, which adopt the matrix-vector iterative method to solve it. We provide empirical experiments on visual multi-point matching task. We first compute a certain node-to-point distance e.g. cosine distance using the learned node feature from graph matching networks (e.g. (Wang et al., 2021))[1] and adopt RCOT-Sinkhorn for the inference in these neural matching model, which shows a great improvement without more training on the features.

Furthermore, based on COT, we discover a side product: a new and easily comprehensible formulation for the Travelling Salesman Problem (TSP), which we refer to as TSP-COT. In this formulation, we construct closed-loop circuits for TSP using the cycle-consistency constraint. To compute an approximate solution for TSP-COT, the regularized approach involves obtaining the approximated probability matrix referred to as a heatmap in the research literature related to TSP. We utilize a post-process search method (i.e. Monte Carlo Tree Search (MCTS) (Fu et al., 2021)) on the heatmap, leading to competitive results. In conclusion, we make the following contributions:

1) We generalize OT to the marginal consistent case called Consistent Optimal Transport (COT), which solves multiple transportation mappings between measures while trying to ensure cycle-consistency among these mappings. Both its Monge and Kantorivich matching formulations are developed.

2) We model the COT problem by adopting entropic and cycle-consistency regularization, and propose an iterative approximate Sinkhorn algorithm named RCOT-Sinkhorn. We apply the RCOT-Sinkhorn algorithm to multi-point matching task and the competitive experimental results show the superiority of our methods.

3) We introduce a new formulation for the Travelling Salesman Problem called TSP-COT, in which we incorporate cycle-consistency to capture the loop constraint. We use the regularized TSP-COT formulation to compute the probability matrix of TSP for efficient search and the post-process search method (MCTS (Fu et al., 2021)) is applied to get the solution. The experiment shows the competitive results of our method.

## 2 RELATED WORKS AND PRELIMINARIES

**Optimal Transportation.** Given two probability measures $\alpha$ and $\beta$ supported on $\mathcal{X}$ and $\mathcal{Y}$, the Monge formulation of Optimal Transportation (Monge, 1781) aims to find a mapping $T : \mathcal{X} \to \mathcal{Y}$ that minimizes:

$$\min_T \{ \int_{\mathcal{X}} c(x, T(x)) d\alpha(x) : T_\# \alpha = \beta \} \tag{1}$$

where $c(\cdot, \cdot)$ is the cost function and the push-forward measure $\beta = T_\# \alpha$ means the satisfaction $\beta(\mathcal{S}) = \alpha(x \in \mathcal{X} : T(x) \in \mathcal{S})$, for an arbitrary set $\mathcal{S} \subset \mathcal{Y}$. The Monge problem is exactly not easy to calculate and an optimal T might not exists, and a popular improvement is the Kantorovich relaxation (Kantorovich, 1942) which seeks the coupling $\mathbf{P}$ instead. Specifically, for the discrete case, we assume $\alpha = \sum_{i=1}^n \mathbf{a}_i \delta_{x_i}$ and $\beta = \sum_{j=1}^m \mathbf{b}_j \delta_{y_j}$ where $(\{x_i\}, \{y_j\})$ are the locations from $(\mathcal{X}, \mathcal{Y})$, and $(\mathbf{a}, \mathbf{b})$ are probability vectors. Then the Kantorovich problem finds the coupling $\mathbf{P}$, specified as

$$\min_{\mathbf{P} \in U(\mathbf{a}, \mathbf{b})} \langle \mathbf{C}, \mathbf{P} \rangle = \sum_{ij} \mathbf{C}_{ij} \mathbf{P}_{ij}, \tag{2}$$

where $U(\mathbf{a}, \mathbf{b}) = \{ \mathbf{P} \in R_{nm}^+ | \mathbf{P} \mathbf{1}_m = \mathbf{a}, \mathbf{P}^\top \mathbf{1}_n = \mathbf{b} \}$ and $\mathbf{C}$ is the cost matrix defined by the divergence between $\{x_i\}_{i=1}^n$ and $\{y_j\}_{j=1}^m$. This minimization can link to the linear program (Bertsimas & Tsitsiklis, 1997) but the calculation speed is really slow for high dimensions. Entropic regularization (Cuturi, 2013) is one of the simple yet efficient methods for solving OT problems:

$$\min_{\mathbf{P} \in U(\mathbf{a}, \mathbf{b})} \langle \mathbf{C}, \mathbf{P} \rangle - \epsilon H(\mathbf{P}), \tag{3}$$

where the entropic regularization $H(\mathbf{P}) = -\langle \mathbf{P}, \log \mathbf{P} - \mathbf{1}_{m \times n} \rangle$. Note $\epsilon > 0$ is the regularization coefficient. It can be solved by Sinkhorn iterations by vector-matrix multiplication (Cuturi, 2013).

---

[1]They embed the structure into node features hence the output is node-wise features suitable in our setting.

**Multi-marginal Optimal Transport.** (MMOT) Instead of coupling two histograms $(\mathbf{a}, \mathbf{b})$ in Kantorovich problem, the multi-marginal optimal transportation couples $K$ histograms $(\mathbf{a}^k)_{k=1}^K$ by solving the following multi-marginal transport (Abraham et al., 2017):

$$\min_{\mathbf{P} \in U((\mathbf{a}^k)_k)} \langle \mathbf{C}, \mathbf{P} \rangle = \sum_k \sum_{i_k=1}^{n_k} \mathbf{C}_{i_1, i_2, \dots, i_K} \mathbf{P}_{i_1, i_2, \dots, i_K} \tag{4}$$

where $\mathbf{C}_{i_1, i_2, \dots, i_K}$ is $n_1 \times \cdots \times n_K$ cost tensor and the valid coupling set $U((\mathbf{a}^k)_k)$ is defined as

$$\{\mathbf{P} \in \mathbb{R}_{n_1 \times n_2 \dots n_K}^+ | \forall k, \forall i_k, \sum_{l \neq k} \sum_{i_l=1}^{n_l} \mathbf{P}_{i_1, \dots, i_K} = \mathbf{a}_{i_k}^k\}. \tag{5}$$

MMOT and our COT both deal with multiple distributions. However, the multi-marginal OT primarily emphasizes learning the joint coupling among more than two distributions, whereas our focus is on learning the coupling between adjacent pairs of a series of distributions and maintaining cycle-consistency constraints among these couplings. And MMOT is a generalized form that indeed presents difficulties when it comes to solving specific problems. When attempting to solve a particular problem using MMOT, it is required to define a specific cost (e.g. P160 in (Peyré et al., 2019)) and thus adds complexity. While the cost in COT is defined between adjacent pairs of a series of distributions and does not necessitate an additional, separate definition.

**Cycle-Consistency for Visual Point Matching** (PM) (Sarlin et al., 2020; Sun et al., 2021). The idea of cycle consistency is widely considered in learning and vision. For example, it is applied in multiple graph matching (Wang et al., 2021; Bernard et al., 2019; Tourani et al., 2023), image matching (Sun et al., 2023; Bernard et al., 2019), and shape matching (Bhatia et al., 2023; Bernard et al., 2019). These multiple matching instances with cycle-consistency in various fields motivate us to investigate whether multiple transportation can be performed with cycle-consistent constraints in the Optimal Transport problem. Thus, in this paper, we elaborate on the concept of cycle-consistency in OT and introduce the definition of COT in Sec.3.1. Visual PM is a prominant area in vision that aims to find optimal point correspondences between images, with wide applications, such as 3D structure estimation and camera pose estimation. Graph matching (GM) (Caetano et al., 2009) builds upon PM and treats the point sets as graphs, aiming to find the optimal node correspondences between graph-structured data. GM can be typically formulated as Lawler's Quadratic Assignment Problem (LAP) (Crama & Spieksma, 1992), which is known to be NP-hard and requires expensive and complex solvers. Recent works (Wang et al., 2019; Yu et al., 2019) have focused on learning node features using supervised or unsupervised loss functions. In this paper, our main focus is on multi-point matching (Swoboda et al., 2019), where we utilize the trained models (Wang et al., 2019) to extract point features and perform inference on testing data, which emphasizes the cycle-consistency among multiple images, enabling more robust and accurate matching results. Traditional methods apply cycle-consistency only to the model's loss function to enhance feature learning on the training set, while not utilizing cycle-consistency during inference on the test set. In contrast, our method employs a training-free approach that assumes consistency is satisfied on the test set, using this prior information to improve performance during inference.

**Travelling Salesman Problem**. In recent years, there has been a surge of interest in leveraging machine learning techniques to address the Travelling Salesman Problem (TSP). The most advanced state-of-the-art methods do not generate the solution directly but instead output a heatmap to indicate the probability of an edge being part of the ground truth routes. Various search methods are utilized to obtain the final solution. The approaches for heatmap (i.e. probability matrix for routing) generation can be categorized into supervised learning, unsupervised learning, and reinforcement learning. Supervised methods, such as GCN (Joshi et al., 2019) and ATT-GCN (Fu et al., 2021), utilize labeled TSP instances to generate heatmaps. Similarly, DIFUSCO (Sun & Yang, 2023) employs diffusion models for heatmap construction, while T2TCO (Li et al., 2024) enhances these maps using learned distributions for gradient-based searches. Unsupervised learning methods, like UTSP (Min et al., 2024), train models without explicit labels, focusing on identifying Hamiltonian cycles through Scattering Attention Graph Neural Networks (SAGs). Reinforcement learning strategies, including those advanced by DIMES (Qiu et al., 2022), optimize sampling efficiency within reinforcement frameworks. As for search methods based on generated heatmaps, greedy algorithms remain prevalent, ranking edges based on their probability scores and adding them iteratively without causing conflicts (Graikos et al., 2022). Monte Carlo Tree Search (MCTS) (Fu et al., 2021), known for its robustness,

simulates multiple scenarios to refine the path continuously. Additionally, local search techniques like 2-opt (Croes, 1958) offer further refinements by swapping edge pairs to improve route efficiency. Amidst this landscape, (Xia et al., 2024) proposed SoftDist, a method that stands out due to its simplicity and effectiveness. By applying the softmax function to the distance matrix and MCTS as the post-process method, SoftDist generates competitive results compared with many complex ML models. In this paper, we attempt to improve SoftDist from an optimization perspective, which can be viewed as an entropic regularized Optimal Transport with row normalization constraints. And building upon COT, a new optimization-based matrix iteration algorithm is proposed to compute a new heatmap for enhancing the simple SoftDist.

## 3 CONSISTENT OPTIMAL TRANSPORT

We begin by presenting the Monge and Kantorovich formulations for Consistent Optimal Transport (COT) in Sec. 3.1. Then, we introduce regularized terms as incorporated into COT, leading to the development of the iterative Sinkhorn algorithm (called RCOT-Sinkhorn Algorithm) in Sec. 3.2. Lastly, in Sec. 3.3, we leverage the principles of COT to devise a novel formulation of the Travelling Salesman Problem (TSP) that highlights the theoretical potential of COT.

### 3.1 THE MONGE AND KANTOROVICH FORMULATION OF COT

**COT's Monge Formulation.** We first assume $K$ probability measures $(\alpha_k)_{k=1}^K$ supported on the space $(\mathcal{X}_k)_{k=1}^K$. For simplicity, we define that $\mathcal{X}_{K+1} = \mathcal{X}_1$ and $\alpha_{K+1} = \alpha_1$. Then COT aims to find mappings $(T_k)_{k=1}^K$ where $T_k : \mathcal{X}_k \to \mathcal{X}_{k+1}$ by optimizing the objective function, i.e.,

$$\min_{(T_k)_k \in \mathcal{C}((\alpha_k)_k)} \sum_k \int_{\mathcal{X}_k} c_k(x, T_k(x)) d\alpha_k(x), \tag{6}$$

where $c_k(\cdot, \cdot)$ is the cost function for the space $(\mathcal{X}_k, \mathcal{X}_{k+1})$. The constraint $\mathcal{C}((\alpha_k)_k)$ is specified:

$$\mathcal{C}((\alpha_k)_k) = \{(T_k)_{k=1}^K | (T_k)_{\#} \alpha_k = \alpha_{k+1}, \forall k; T_K \circ T_{K-1} \circ \cdots \circ T_2 \circ T_1(X) = X, \forall X \subset \mathcal{X}_1\}, \tag{7}$$

where $(T_k)_{\#} \alpha_k = \alpha_{k+1}$ is the push-forward operation from measure $\alpha_k$ to $\alpha_{k+1}$ satisfying $\alpha_{k+1}(B \in \mathcal{X}_{k+1}) = \alpha_k(x \in \mathcal{X}_k | T_k(x) \in B)$ for any measurable set $B$. And $\forall X \subset \mathcal{X}_1$, the equality $T_K \circ T_{K-1} \circ \ldots T_2 \circ T_1(X) = X$ is the cycle-consistency constraint that enforces the final transport results aligning to the original one beginning at points in $\mathcal{X}_1$. Naturally, we can get the measure $\alpha_1(X) = \alpha_1(T_K \circ \cdots \circ T_1(X))$. Note the cycle-consistency starts from $\alpha_1$ and one can also formulate the COT's Monge problem starting from $\alpha_2, \alpha_3, \ldots, \alpha_K$. For the calculation, the COT's Monge formulation encounters difficulties like those of traditional Monge OT and the solution may even not exist in discrete cases. Hence for COT, we only consider the discrete case that various points in different domain need to get the matching with cycle-consistency constraints.

**COT's Kantorovich Relaxation for Matching.** Here, we discuss the discrete case of COT, where there are $N$ points to be matched in each space $(\mathcal{X}_k)_{k=1}^K$, ensuring that the matching satisfies cycle-consistency constraints. Let's assume that the measures are represented by $\alpha_k = \sum_{i=1}^N \delta_{x_i^k}$, where $x_i^k$ denotes the location of the $i$-th point in the space $\mathcal{X}_k$. In the multi-matching scenario, the goal of COT is to find $K$ doubly stochastic matrices $(\mathbf{P}_k)_{k=1}^K$, where $\mathbf{P}_k$ is the coupling matrix that satisfies:

$$\min_{(\mathbf{P}_k)_k} \sum_{k=1}^K \langle \mathbf{C}_k, \mathbf{P}_k \rangle, \text{ s.t. } \mathbf{P}_k \mathbf{1}_N = \mathbf{1}_N, \mathbf{P}_k^\top \mathbf{1}_N = \mathbf{1}_N, \prod_{k=1}^K \mathbf{P}_k = \mathbf{I}, \mathbf{P}_k \in \{0,1\}^{N \times N}, \forall k \tag{8}$$

Here, $\mathbf{I}$ represents the identity matrix, and $\mathbf{1}_N$ denotes a column vector with all elements equal to 1. The constraints $\prod_{k=1}^K \mathbf{P}_k = \mathbf{I}$ aim to ensure cycle-consistency. Specifically, given the points in $\mathcal{X}_k$, we consider $\mathbf{P}_k$ as the transition from $\mathcal{X}_k$ to $\mathcal{X}_{k+1}$ with the matrix $\mathbf{P}_k$ satisfying the matching between the points $x_i^k$ and $x_j^{k+1}$ if $(\mathbf{P}_k)_{ij} = 1$. Note that $(\mathbf{P}_k)_{k=1}^K$ are permutation matrices and proof is given in Appendix A. Similarly, with the transition matrix $\mathbf{P}_k \mathbf{P}_{k+1}$, we can know the matching between the points $x_i^k$ and $x_j^{k+2}$ if $(\mathbf{P}_k \mathbf{P}_{k+1})_{ij} = 1$. Thus we can view the transition matrix $\prod_{k=1}^K \mathbf{P}_k$ as the transportation $\mathcal{X}_1$ to $\mathcal{X}_{K+1}$ and $\prod_{k=1}^K \mathbf{P}_k = \mathbf{I}$ is the constraints for cycle-consistency. However, Eq. 8 is no longer a linear programming due to the constraints of cycle-consistency. For efficiency, we propose the regularized COT, which allows to derive an matrix-vector iterative algorithm for obtaining approximate solutions.

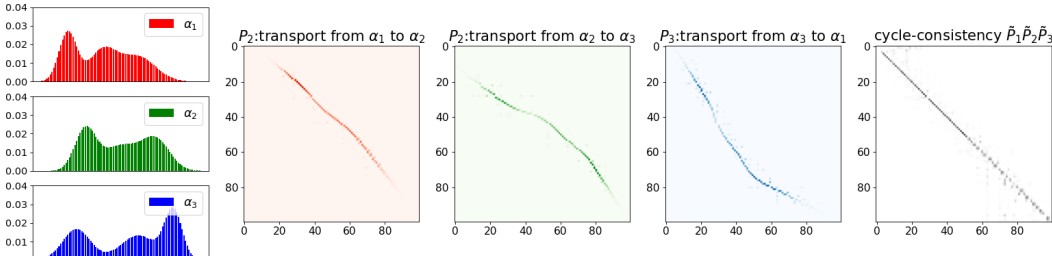

Figure 2: Illustration of transport solutions $\mathbf{P}_1$, $\mathbf{P}_2$, and $\mathbf{P}_3$, along with the cycle-consistency matrix $P_1P_2P_3$, based on three given histograms. The histograms on the left correspond to $\alpha_1$, $\alpha_2$, and $\alpha_3$, while the middle histograms represent the couplings from $\alpha_1$ to $\alpha_2$, $\alpha_2$ to $\alpha_3$, and $\alpha_3$ to $\alpha_1$. Finally, the rightmost matrix denotes $P_1P_2P_3$, which exhibits a close similarity to the identity matrix.

### 3.2 Regularized COT and The RCOT-Sinkhorn Algorithm

Due to the constraint $\prod_k \mathbf{P}_k = \mathbf{I}$, a straightforward idea is to relax the optimization using a regularization term. This can be achieved by adopting the following optimization:

$$\min_{\forall k, \mathbf{P}_k \in U(\mathbf{a}^k, \mathbf{a}^{k+1})} \mathcal{E}_{\text{RCOT}} = \sum_{k=1}^{K} \langle \mathbf{C}_k, \mathbf{P}_k \rangle + \delta' \mathcal{D}(\mathbf{P}_k, \mathbf{I}), \tag{9}$$

where $\mathcal{D}(\mathbf{P}_k, I) = ||\mathbf{P}_k - \mathbf{I}||_F^2$ and $\delta' > 0$ is the coefficient for cycle-consistent regularization. It can be observed that this optimization problem, similar to solving the Gromov-Wasserstein (GW) Distance, is a non-convex optimization. Taking inspiration from the methods used to solve the Gromov-Wasserstein Distance, we further regularize the optimization using entropy regularization as follows:

$$\min_{\forall k, \mathbf{P}_k \in U(\mathbf{a}^k, \mathbf{a}^{k+1})} \mathcal{E}_{\text{RCOT}} - \epsilon \sum_k H(\mathbf{P}_k), \tag{10}$$

where $H(\cdot)$ is the entropic regularization with its cofficent $\epsilon$. Two algorithms, namely RCOT-Sinkhorn and RCOT-PGD, are proposed to solve this problem, which are defined in Algorithm 1 and Algorithm 2.

**RCOT-Sinkhorn Algorithm.** Following the algorithms proposed for approximating the computation of GW in (Peyré et al., 2016), we use iteratively Sinkhorn's algorithm to progressively compute a stationary point of Eq. 10. Indeed, successive linearizations of the objective function lead to consider the succession of updates

$$\mathbf{P}_k^{(l+1)} = \arg \min_{\mathbf{P}_k \in U(\mathbf{a}^k, \mathbf{a}^{k+1})} \langle \mathbf{C}_k^{(l)}, \mathbf{P}_k \rangle - \epsilon H(\mathbf{P}_k). \tag{11}$$

Note $\mathbf{C}_k^{(l)} = \nabla \mathcal{E}_{\text{RCOT}}(\mathbf{P}_k^{(l)}) = \mathbf{C}_k - \delta' \mathbf{M}_k^{(l)}$ where $\mathbf{M}_k^{(l)}$ is

$$\text{Diag}\left(\frac{1}{\mathbf{a}^k}\right) \left(\prod_{t_1=1}^{k-1} \mathbf{P}_{t_1}^{(l)}\right)^\top \left(\prod_{t_2=1}^{K} \mathbf{P}_{t_2}^{(l)} - \mathbf{I}\right) \left(\prod_{t_3=k+1}^{K} \mathbf{P}_{t_3}^{(l)}\right)^\top. \tag{12}$$

Note Eq. 11 can be solved by Sinkhorn Algorithm, thus we can adopt the Sinkhorn iteratively for the solution of RCOT. More proof details are given in Appendix B. Note the above iterations can be interpreted as a mirror-descent scheme, in which the convergence is discussed in (Aubin-Frankowski et al., 2022; Zhou et al., 2020) When $\delta' = 0$, the solution $\mathbf{P}_k$ degenerates into the vanilla entropic OT and when $\delta' > 0$, $(\mathbf{P}_k)_k$ tend to satisfy cycle-consistency. Note our RCOT-Sinkhorn IS presented in Appendix B. As shown in Fig. 3, 6 points are sampled from three 2D-Gaussian distributions and Euclidean distances are used as costs for computing couplings. Compared to the pair-wise Sinkhorn algorithm, our RCOT-Sinkhorn achieves cycle-consistency results. Fig. 2 illustrates the transportation results among more complex distributions. It is noteworthy that the left three histograms are sampled from Gaussian mixture distributions, and the couplings can be computed using the RCOT-Sinkhorn algorithm as shown in the middle three subfigures. As shown in the rightmost subfigure, the cycle-consistency $\prod_{k=1}^{K} \mathbf{P}_k = \mathbf{I}$ is almost satisfied.

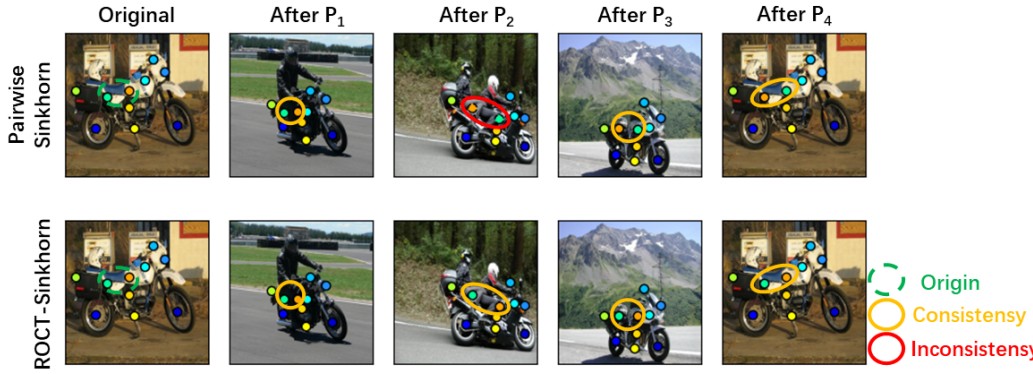

Figure 4: Example point matching results after applying $\mathbf{P}_1$, $\mathbf{P}_2$, $\mathbf{P}_3$ and $\mathbf{P}_4$ to the original point set. The features are extracted using NMGM (Wang et al., 2021). We observe that the Sinkhorn method fails to achieve cycle-consistency, while our RCOT-Sinkhorn method successfully maintains cycle-consistency, resulting in the graph after applying $\mathbf{P}_2$ remaining identical to the original graph.

**Applications in Multi-point Matching.** Here, we apply the RCOT to the inference of the multi-point matching model. We assume the existence of multiple sets, each containing several point features extracted from images by the trained neural model. Our goal is to establish cycle-consistent matches among these sets. Specifically, given $K$ probability measures $(\alpha_k)_{k=1}^{K}$, where $\alpha_k = \sum_{i=1}^{N} \delta'_{x_i^k}$, and $x_i^k$ represents the point feature, we can define the cost $\mathbf{C}_k$ between $\alpha_k$ and $\alpha_{k+1}$. Then the inference during the testing process can be formulated with Eq. 10. To solve the optimization, we utilize the RCOT-Sinkhorn algorithm to obtain predictions for the testing data. Fig. 4 illustrates the inference re-

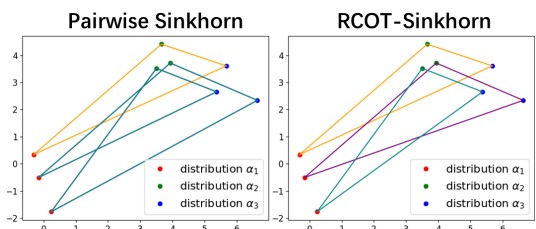

Figure 3: Multi-matching on 2-D points. Left: result of pairly adopting the Sinkohrn algorithm; Right: solutions of our RCOT-Sinkhorn Algorithm 1. Our matching forms a closed loop, whereas the pairwise Sinkhorn results do not.

sults using Pairwise Sinkhorn and RCOT-Sinkhorn algorithms, where a neural matching model (NMGM (Wang et al., 2021)) serves as the backbone. It can be observed that the coupling $\mathbf{P}_2$ generated by the pairwise Sinkhorn method contains a mismatch for two points at the rear of the vehicle. However, our algorithms correct this misalignment and produce accurate matching. One direct concern for multiple matching is the order of point sets in the matching process, as different matching orders may affect the prediction results of RCOT-Sinkhorn or RCOT-PGD. In practice, for $K = 3$, different selection orders are theoretically equivalent for RCOT. However, for $K > 3$, different selection orders can indeed theoretically affect the prediction results of the RCOT algorithm as shown in Tab. 5, Tab. 1 and Fig. 7. Nevertheless, based on existing experiments, it seems that changing the order has little impact on the prediction results.

**The setting of Hyper-parameters $\delta'$ and $\epsilon$.** The traditional method of tuning hyper-parameters is grid search, but this approach has a large computational cost and low efficiency. In situations where the model itself has high computational requirements or when dealing with large-scale data, the feasibility of using grid search is limited. Inspired by binary search, we proposed the method for tuning the hyper-parameter in Algorithm 5. This method can achieve logarithmic convergence speed, significantly reducing the computational load of tuning parameters.

### 3.3 A NEW COT-BASED TSP FORMULATION AND A HEATMAP-BASED SOLVING METHOD

From the multiple transportation view, we consider what if all the transportation is in the same space. This means we can set that $\alpha = \alpha_1 = \alpha_2 = \cdots = \alpha_K$ and then all probability measures share the same locations and histograms, which leads to the same cost matrix (i.e. $\mathbf{C} = \mathbf{C}_1 = \mathbf{C}_2 = \cdots = \mathbf{C}_K$) and coupling solutions (i.e. $\mathbf{P} = \mathbf{P}_1 = \mathbf{P}_2 = \cdots = \mathbf{P}_K$) for all transportation. Under this

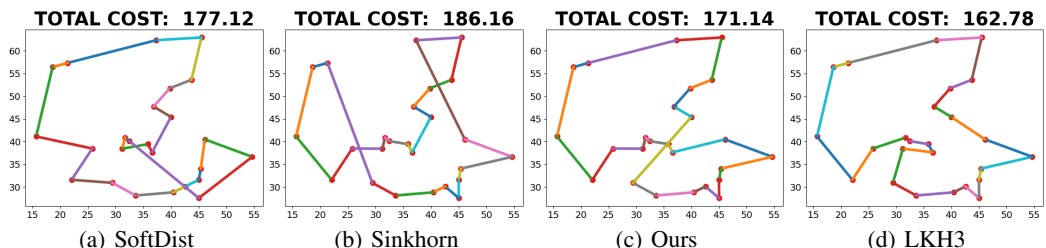

Figure 5: Comparisons on TSP. The left three adopt the greedy method on distance matrix, and probability matrix with Sinkhorn and our Algorithm 2, respectively. The rightmost one is the strong TSP solver LKH3 (Helsgaun, 2017). Our method performs competitively for heatmap generation.

assumption, the cycle-consistency is transformed into $\mathbf{P}^K = \mathbf{I}$, which implies that each point returns to its original location after transportation. This inspires us to draw a connection to TSP.

TSP involves $K$ points with its distance matrix $\mathbf{C}$ to get the solution $\mathbf{P} \in \{0,1\}^{K \times K}$. Note we assume the transport for points themselves is not allowed (i.e. $(\mathbf{C})_{ii} \to \infty$ and thus $(\mathbf{P})_{ii} = 0$). We apply the **cycle-consistency** view in COT to capture the **closed-loop** constraint, which forms a new formulation (TSP-COT):

$$\min_{\mathbf{P} \in \{0,1\}^{K \times K}} \langle \mathbf{C}, \mathbf{P} \rangle \text{ s.t. } \mathbf{P}\mathbf{1}_K = \mathbf{1}_K, \ \mathbf{P}^\top \mathbf{1} = \mathbf{1}, \ \langle \mathbf{P}^k, \mathbf{I} \rangle = 0 (\forall k < K), \ \mathbf{P}^K = \mathbf{I}. \quad (13)$$

Note $\mathbf{P}^k$ is the $k$-th power of matrix $\mathbf{P}$ and the condition $\langle \mathbf{P}^k, \mathbf{I} \rangle = 0$ for $k < K$ is used to terminate the consistency process before the final step, to ensure $(\mathbf{P}^k)_{ii} = 0$, which guarantees that the probability of a travelling salesman starting from position $i$, taking $k$ steps ($k < K$), and returning to position $i$ is zero. On the other hand, the condition $(\mathbf{P})^K = \mathbf{I}$ is imposed to enforce cycle-consistency, ensuring that $(\mathbf{P}^K)_{ii} = 1$. It guarantees that the salesman returns to start. The optimization in Eq. 13 is no longer a Linear Program problem. Similar to that in Sec. 3.2, the entropic and closed-loop regularization is employed for minimization:

$$\min_{\mathbf{P} \geq \mathbf{0}} \langle \mathbf{C}, \mathbf{P} \rangle - \epsilon H(\mathbf{P}) + \sum_k \delta'_k ||\mathbf{P}^k - \mathbf{I}||_F^2 \quad \text{s.t.} \quad \mathbf{P}\mathbf{1}_K = \mathbf{1}_K, \ \mathbf{P}^\top \mathbf{1}_K = \mathbf{1}_K, \quad (14)$$

where $(\delta'_k)_k$ are the regularization coefficients. We set $\delta'_k < 0$ for $k < K$ to make $(\mathbf{P}^k)_{ii}$ approach 0 for every $k < K$, and $\delta'_K > 0$ to make $(\mathbf{P}^k)_{ii}$ approach 1. Note that Eq. 14 and the Gromov-Wasserstein problem are both non-convex optimizations. To handle this, we employ linearizations of the objective function, which allow us to consider updates

$$\mathbf{P}^{(l+1)} = \arg \min_{\mathbf{P} \in U(\mathbf{1}_K, \mathbf{1}_K)} \langle \mathbf{C}^{(l)}, \mathbf{P} \rangle - \epsilon H(\mathbf{P}), \quad (15)$$

where

$$\mathbf{C}^{(l)} = \mathbf{C} + \mathbf{M}, \quad \mathbf{M} = \sum_{k=1}^{K} \sum_{t=0}^{k-1} 2\delta'_k (\mathbf{P}^t)^\top (\mathbf{P}^k - \mathbf{I})(\mathbf{P}^{k-1-t})^\top. \quad (16)$$

Then we can obtain the approximate solution of TSP with the iterative Sinkhorn algorithm as proposed in Algorithm 3 and details are given in Appendix C. However, unfortunately, Algorithm 3 can not achieve the ideal closed-loop solution which may be due to the simple setting of $\delta'_k$ and too many regularized terms of closed-loop constraints.

In fact, our probability-based results on the other hand enable the selection of the TSP path from a probabilistic perspective rather than relying solely on the traditional distance matrix. This shift transforms TSP into a sampling problem, where the calculated probability matrix can be utilized. For example, we can employ a greedy method, as described in Appendix C, to search for a closed-loop path based on the probability matrix computed using Algorithm 2. In Fig. 5, 25 points are randomly sampled as the locations and we compare the total cost based on greedy search using the Euclidean distance matrix, Sinkhorn probability matrix, and our probability matrix calculated by Algorithm 2 in Fig. 5 and find that our approach performs competitively.

Though our current method is still far from competing with strong classic TSP solvers like LKH3 (Helsgaun, 2017), as shown in Fig. 5, it provides a new perspective for tackling the TSP, which

Table 1: **Comparison of different methods on Willow and Pascal VOC with Berkeley annotations.** Accuracy (ACC), Consistent rate (CR) and Consistent Accuracy (CACC) (%) are reported and our RCOT-Sinkhorn or RCOT-PGD algorithms outperform in ACC, CR, and CACC evaluations.

| Data | Methods | Three Point Sets Matching | | | Four Point Sets Matching | | |
|---|---|---|---|---|---|---|---|
| | | ACC | CR | CACC | ACC | CR | CACC |
| Willow | NMGM | 91.02 | 93.76 | 84.28 | 92.74 | 96.34 | 84.44 |
| | **NMGM-COT (ours)** | **92.30** | **99.72** | **88.28** | **93.04** | **99.78** | **85.90** |
| | PCA-GM | 92.64 | 90.12 | 84.86 | 93.04 | 87.16 | 81.00 |
| | **PCA-GM-COT (ours)** | **93.27** | **98.96** | **90.04** | **93.22** | **96.16** | **85.52** |
| | IPCA-GM | 94.51 | 91.44 | 87.76 | 94.08 | 87.38 | 82.21 |
| | **IPCA-GM-COT (ours)** | **95.06** | **95.00** | **90.31** | **94.60** | **92.22** | **85.57** |
| | CIE-H | 93.08 | 84.72 | 82.88 | 93.04 | 80.86 | 78.48 |
| | **CIE-H-COT (ours)** | **94.62** | **98.47** | **91.48** | **94.24** | **97.65** | **88.07** |
| VOC | PCA-GM | 68.14 | 66.64 | 45.28 | 70.49 | 61.31 | 40.53 |
| | **PCA-GM-COT (ours)** | **68.74** | **82.69** | **51.63** | **70.60** | **63.74** | **41.54** |
| | IPCA-GM | 69.83 | 69.02 | 48.94 | 71.33 | 63.92 | 43.92 |
| | **IPCA-GM-COT (ours)** | **70.93** | **84.68** | **55.24** | **71.86** | **78.99** | **50.01** |
| | CIE-H | 72.67 | 68.54 | 51.04 | 74.53 | 63.74 | 45.78 |
| | **CIE-H-COT (ours)** | **73.52** | **83.57** | **57.65** | **75.40** | **76.41** | **51.60** |

involves converting the distance matrix into a probability matrix and searching for the optimal path based on the probabilities[2]. In the probability matrix, the edge selection is based on global considerations, which inherently provides an advantage over distance-based edge selection. Moreover, if an improved algorithm can be developed to obtain the closed-loop probability solution in the future, we would no longer need the sampling algorithm to determine the TSP path. This opens up possibilities for efficiently solving large-scale TSP problems using matrix scaling methods via GPU computing.

**Time Complexity Analysis.** The time complexity of 3 is primarily reflected in Eq. 16, which involves two nested loops, running $O(K^2)$ times. Inside each loop, there are operations involving matrix exponentiation and matrix multiplication of order $K$. We can compute the matrix powers using eigenvalue decomposition, which reduces the time complexity of computing matrix powers within the loop to $O(K^3)$, which is the same as matrix multiplication. Therefore, the overall time complexity of the algorithm is $O(K^5)$. Although our algorithm has a high time complexity, it is a training-free approach. In comparison to retraining neural network parameters, our method holds certain advantages.

Further discussions about COT are shown in Appendix H.

## 4 EXPERIMENTS

### 4.1 EXPERIMENTS ON VISUAL POINT MATCHING ACROSS SETS

We evaluate the task of keypoint matching on Pascal VOC dataset with Berkeley annotations (Everingham et al., 2010; Bourdev & Malik, 2009) and Willow Object Class dataset (Cho et al., 2013). For the evaluation metric, the average accuracy (ACC) (Wang et al., 2021) can be regarded with consistency matching view between two measures:

$$\text{ACC} = \frac{1}{N} \langle \prod_{k=1}^{2} \hat{\mathbf{P}}_k \odot \mathbf{Y}_k, \mathbf{I}_N \rangle \tag{17}$$

where $\hat{\mathbf{P}}_1 = \hat{\mathbf{P}}_2^\top$ is the one-hot matching prediction of $\mathbf{P}_1$. $\mathbf{Y}_k$ is the ground truth for $\mathbf{P}_k$ and $\odot$ denotes element-wise matrix multiplication. Then we can extend the two measure evaluation to more measure case. To evaluate matching results, we develop two metrics called **Consistent Rate (CR)**

---

[2]We believe that there is a potential of adapting our techniques to more combinatorial problems beyond TSP.

Table 2: Comparison of different methods on TSP-200/500/750 datasets. The average tour length (Length), the average performance gap (Gap) between the solution generated by Concorde and the average inference time (Time) are reported. Baseline results are sourced from (Fu et al., 2021; Qiu et al., 2022; Xin et al., 2021)

| Method | Type | TSP-200 | | | TSP-500 | | | TSP-750 | | |
|---|---|---|---|---|---|---|---|---|---|---|
| | | Length | Gap | Time | Length | Gap | Time | Length | Gap | Time |
| Concorde | Exact | 10.72 | 0.00% | 17.86M | 16.58 | 0.00% | 37.66M | 20.09 | 0.00% | 4.35H |
| LKH-3 | Heuristic | 10.72 | 0.00% | 46.28M | - | - | - | 20.09 | 0.00% | 2.57H |
| T2T | SL+G | 10.85 | 1.38% | 3.45M | - | - | - | 20.85 | 3.81% | 20.03M |
| T2T | SL+G+2opt | 10.75 | 0.47% | 5.16M | - | - | - | **20.23** | **1.13%** | 18.83M |
| T2T | SL+MCTS | **10.72** | **0.10%** | 1.28H | - | - | - | 20.71 | 3.08% | 14.55H |
| AM | RL+S | - | - | - | 22.64 | 38.84% | 15.64M | - | - | - |
| AM | RL+G | - | - | - | 20.02 | 20.99% | **1.51M** | - | - | - |
| AM | RL+BS | - | - | - | 19.53 | 18.03% | 21.99M | - | - | - |
| GNN | SL+G | 11.95 | 11.72% | 1.23M | - | - | - | 23.12 | 15.58% | **1.50M** |
| GNN | SL+G+2opt | 10.90 | 1.80% | 1.53M | - | - | - | 20.41 | 2.05% | 2.12M |
| GNN | SL+MCTS | 10.73 | 0.30% | 0.47H | - | - | - | 21.71 | 8.08% | 3.61H |
| DIMES | RL+G | 12.45 | 16.38% | 1.37M | - | - | - | 23.36 | 16.26% | 1.59M |
| DIMES | RL+MCTS | 11.16 | 4.29% | **0.52M** | **16.84** | **1.77%** | 2.64M | 20.58 | 2.44% | 5.42M |
| SoftDist | MCTS | 10.74 | 0.25% | **1.23M** | 16.80 | 1.32% | **1.67M** | 20.57 | 1.95% | **3.7M** |
| COT (ours) | MCTS | **10.74** | **0.23%** | 1.28M | **16.78** | **1.21%** | 2.10M | **20.47** | **1.92%** | 6.24M |

and **Consistent Accuracy (CACC)** to assess the cycle-consistency effectiveness of the inference method. These metrics are defined as follows:

$$\text{CR} = \frac{1}{N}\langle \prod_{k=1}^{K} \hat{\mathbf{P}}_k, \mathbf{I}_N \rangle, \quad \text{CACC} = \frac{1}{N}\langle \prod_{k=1}^{K} (\hat{\mathbf{P}}_k \odot \mathbf{Y}_k), \mathbf{I}_N \rangle \tag{18}$$

where $\hat{\mathbf{P}}_k$ is the one-hot matching prediction of $\mathbf{P}_k$ for the $k-$th point set to $(k+1)-$th set. Note **CR** refers to the accuracy of forming cycles via matching, while **CACC** represents the accuracy of forming cycles where each feature point within the cycle is matched correctly.

**Results.** The results of visual matching task are summarized in Tab. 1. We use the previous neural matching models, namely NMGM (Wang et al., 2021), PCA-GM (Wang et al., 2019), IPCA-GM (Wang et al., 2020) and CIE-H (Yu et al., 2019)), as the backbone to evaluate our inference algorithm. We compare RCOT-Sinkhorn and RCOT-PGD with (Munkres, 1957), EMD (Dantzig, 1949) and Sinkhorn Algorithm (Cuturi, 2013). As shown in Tab. 1, for the Willow dataset, our methods outperform all other inference methods by ACC, CR and CACC evaluations. For the experiments on Pascal VOC with Berkeley annotations, our RCOT-Sinkhorn outperforms others across all backbones. Results for multiple (more than three) measures and order switching and the details of hyper-parameters and running time are discussed in Tab 1.

## 4.2 EXPERIMENTS ON TRAVELLING SALESMAN PROBLEMS

We evaluate our algorithm on the Travelling Salesman Problem (TSP) using the TSP-200/500/750 datasets. Each dataset includes 128 instances. TSP instances are generated by sampling N nodes uniformly from the unit square.

**Evaluation Metrics and Baselines.** We report the average tour length (Length), average performance gap (Gap), and average inference latency time (Time). Length (lower is better) represents the average length of the predicted tour for each graph in the test set. Gap (smaller is better) measures the average relative performance gap in solution length compared to a baseline method. Time (shorter is better) denotes the average clock time to generate solutions for each test instance, reported in seconds (s), minutes (m), or hours (h). As for baselines, we evaluate Concorde (Applegate et al., 2006), T2T (Li et al., 2024), GNN (Joshi et al., 2019), DIMES (Qiu et al., 2022), SoftDist (Xia et al., 2024), etc. Based on the characteristics of these methods, we categorize them into three types: Solvers, ML-based methods and Training-free methods.

**Results.** The results of TSP experiments are summarized in Tab. 2. Without additional training, our algorithm can achieve performance superior to other models. Specifically, our COT algorithm achieves a performance gap of 0.23% on TSP-200, 1.21% on TSP-500, and 1.92% on TSP-750 with

only a slight increase in the heatmap generation time compared to SoftDist. Our method, compared to DIMES, has clear advantages in terms of the quality of the optimal solutions. And as the problem size increases, the gap in solving time between our method and the DIMES method is also narrowing. Also, our method shares all the merits of SoftDist as mentioned by (Xia et al., 2024): compared with learning-based methods, our method do not need a large labeled training dataset and training time to make the methods work. Nonlearning-based methods can be easily applied to any dataset and consistently produce acceptable routing paths.

**Numerical Convergence Analysis.** We validate the convergence of Alg. 3 through experiments on the TSP-200 dataset with log-log-plot introduced in (Cortes et al., 1993). We plot a log-log graph with the x-axis representing the number of iterations and the y-axis representing the norm difference between the updated and old matrices M, $||M_{old} - M_{new}||_F^2$, at each iteration, and plot the first-order and second-order convergence line (convergence order and convergence line are defined in Appendix F) on the same graph. As shown in Fig. 6, our convergence line (blue) is approximately parallel to the first-order convergence line (orange), indicating that our algorithm converges at the first order. As for the analytic convengence, our algorithm is equivalent to the projection of gradient descent algorithm. The convergence proof related to the our algorithm is discussed in (Peyré et al., 2016).

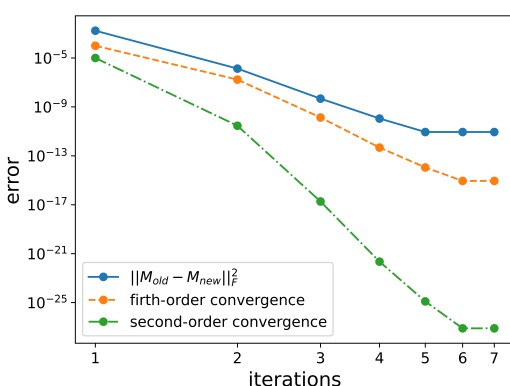

Figure 6: Plot with log scaling on both the x-axis (iterations) and y-axis (the norm difference between the updated and old matrices M in Alg.3 on TSP-200 dataset at each iteration).

### 4.3 MORE EXPERIMENTAL RESULTS

**Experiments on Model Fusion Task.** Following (Singh & Jaggi, 2020) that applies the OT (e.g. Sinkhorn algorithm) for the model fusion task, we apply our RCOT-Sinkhorn algorithm for multi-model fusion. We focus on exploring the benefits of fusing multiple models that only differ in their parameter initializations (i.e., seeds). We study this in the context of networks e.g. MLP and VGG11 which have been trained on MNIST and CIFAR10 respectively. Unlike the previous pairwise Sinkhorn algorithm in (Singh & Jaggi, 2020), we apply our RCOT-Sinkhorn algorithm instead to get the consistency among multiple models. The experiment results are given in Tab.3 and details about our algorithm are given in Appendix D.

**Ablation Study.** We conduct ablation studies to determine the impact of some factors on the effectiveness of our method, like switching the order of point sets, varying $\delta$ and $\epsilon$ in matching process and applying Hungarian algorithm to P in Eq. 16. We switch the order of sets on Willow_3GM ($K = 3$) and Willow_4GM ($K = 4$), and find that the results before and after switching the order given in Tab. 4, 5 are almost the same. We vary $\delta$ and $\epsilon$ in visual matching experiments and the results given in Tab.6 demonstrate the robustness of our method. In the TSP experiments, we attempt to apply the Hungarian algorithm to P in Eq. 16 and find that the final results after tuning are almost the same whether or not the Hungarian algorithm is used, but the optimal parameters are different.

## 5 CONCLUSION, LIMITATION AND FUTURE WORK

We have introduced a generalized form for Consistent Optimal Transport (COT), which enables transportation among multiple measures while (softly) preserving cycle-consistent constraints. Besides, we propose a new formulation of TSP based on TSP-COT, which helps the calculation of heatmap for solving with regularization on TSP-COT. For the limitation, our RCOT-Sinkhorn introduces a hyperparameter $\delta'$ for tuning. Similarly, for the regularized TSP-COT, the number of hyperparameters increases by $K$ i.e. the number of measurements. For future work, we believe that a Schrödinger bridge based on cycle-consistency would be a promising direction, which will be the focus of our next efforts.

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

APPENDIX

## A    PROOF OF $(\mathbf{P}_k)_{k=1}^{K}$ BEING PERMUTATION MATRICES IN EQ. 8

Denote the $(i,j)$ element in matrix $\mathbf{P}_k$ as $p_{ij}^{(k)}$. Suppose the elements of $(\mathbf{P}_k)_{k=1}^{K}$ are in the general range of (0,1). By Cauchy-Schwarz Inequality, the following equation holds and the equality holds if and only if $\exists l^*$ s.t. $p_{il^*}^{(k)} p_{l^* j}^{(k+1)} = 1$ and $p_{il}^{(k)} p_{lj}^{(k+1)} = 0 \ \forall l \neq l^*$.

$$(\mathbf{P}_k \mathbf{P}_{k+1})_{ij} = \sum_{l=1}^{K} p_{il}^{(k)} p_{lj}^{(k+1)} \leq \sum_{k=1}^{K} (p_{il}^{(k)})^2 \sum_{k=1}^{K} (p_{lj}^{(k+1)})^2 \leq 1, k = 1, 2, \ldots, K \qquad (19)$$

If $\mathbf{P}_k$ is not a permutation matrix, the equality in Eq. 19 can not hold and thus the constraints $\prod_{k=1}^{K} \mathbf{P}_k = \mathbf{I}$ can not be satisfied. Thus $(\mathbf{P}_k)_{k=1}^{K}$ are permutation matrices.

## B    LAGRANGIAN METHOD FOR REGULARIZED COT

We first give the derivation of Eq. 8. At first, given the minimization

$$\min_{(\mathbf{P}_k)_k : \mathbf{P}_k \in U(\mathbf{a}^k, \mathbf{a}^{k+1})} \mathcal{L}_1 = \sum_{k=1}^{K} \langle \mathbf{C}_k, \mathbf{P}_k \rangle - \epsilon \sum_k H(\mathbf{P}_k) - \delta' || \prod_{k=1}^{K} \tilde{\mathbf{P}}_k - \mathbf{I} ||_F^2, \qquad (20)$$

we can adopt the Lagrangian method to solve it. For each coupling $\mathbf{P}_k$, we introduce $(\mathbf{f}_k, \mathbf{g}_k)$ to the constraints in $U(\mathbf{a}^k, \mathbf{a}^{k+1})$, i.e.

$$\mathbf{P}_k \mathbf{1}_{n_{k+1}} = \mathbf{a}^k \quad \text{and} \quad (\mathbf{P}_k)^{\top} \mathbf{1}_{n_k} = \mathbf{a}^{k+1}, \qquad (21)$$

and then we can get the Lagrangian function as

$$\mathcal{L} = \mathcal{L}_1 - \sum_k \left( \langle \mathbf{f}_k, \mathbf{P}_k \mathbf{1}_{n_{k+1}} - \mathbf{a}^k \rangle + \langle \mathbf{g}_k, (\mathbf{P}_k)^{\top} \mathbf{1}_{n_k} - \mathbf{a}^{k+1} \rangle \right) \qquad (22)$$

We compute the partial derivative of $\mathcal{L}$ with respect to $\mathbf{P}_k$ as

$$\frac{\partial \mathcal{L}}{\partial \mathbf{P}_k} = \mathbf{C}_k + \epsilon \log \mathbf{P}_k - \mathbf{f}_k \mathbf{1}^{\top} - \mathbf{1} \mathbf{g}_k^{\top} - \delta' M_k = 0, \qquad (23)$$

where $M_k$ is specified as

$$M_k = \frac{\partial f}{\partial \mathbf{P}_k} = \frac{\partial tr(Y^{\top} Y)}{\partial \mathbf{P}_k} = \frac{\partial || \prod_{k=1}^{K} \tilde{\mathbf{P}}_k - \mathbf{I} ||_F^2}{\partial \mathbf{P}_k}. \qquad (24)$$

Here we set $Y = \prod_{k=1}^{K} \tilde{\mathbf{P}}_k - \mathbf{I}$ and $f = tr(Y^{\top} Y)$ in Eq. 24. With the method given in (Hu, 2012), We solve it by utilizing the relationship between matrix derivative and its partial derivatives. Specially, we have

$$df = tr(dY^{\top} Y) + tr(Y^{\top} dY) = tr(2Y^{\top} dY) = tr(\frac{\partial f^{\top}}{\partial Y} dY), \qquad (25)$$

---

**Algorithm 1:** RCOT-Sinkhorn: Iterative Sinkhorn-based Algorithm for Regularized COT

---

**Input:** Cost Matrices $(\mathbf{C}_k)_{k=1}^{K}$ and histograms $(\mathbf{a}^k)_{k=1}^{K}$, iteration number $L$
**Output:** the couplings $(\mathbf{P}_k^{(L-1)})_{k=1}^{K}$
  Initialize $M_k^{(0)} = \mathbf{0}_{n_k, n_{k+1}}$      for all $k$
  **for** $l = 0, 1, \ldots, L-1$ **do**
    **for** $k = 1, 2, \ldots, K$ **do**
      $\mathbf{P}_k^{(l)} = \text{Sinkhorn}(\mathbf{C}_k - \delta' M_k^{(l)}, \mathbf{a}^k, \mathbf{a}^{k+1})$
      $\tilde{\mathbf{P}}_k^{(l)} = \mathbf{P}_k^{(l)} \oslash \mathbf{a}^k$
    **end for**
    Calculate $(M_k^{(l+1)})_k$ by Eq. 12 with $(\tilde{\mathbf{P}}_k^{(l)})_k$
  **end for**

---

---

**Algorithm 2:** RCOT-PGD: Projected Gradient based Algorithm for Regularized COT

---

**Input:** Cost Matrices $(\mathbf{C}_k)_{k=1}^{K}$ and histograms $(\mathbf{a}^k)_{k=1}^{K}$, iteration number $L$
**Output:** the couplings $(\mathbf{P}_k^{(L-1)})_{k=1}^{K}$
  Initialize $M_k^{(0)} = \mathbf{0}_{n_k, n_{k+1}}$      for all $k$
  **for** $l = 0, 1, \ldots, L-1$ **do**
    **for** $k = 1, 2, \ldots, K$ **do**
      $K_{\text{proj}} = \mathbf{P}_{k-1}^{(l)} \odot e^{(-\tau(\mathbf{C}_k - \delta' M_k^{(l)}) + \epsilon \log \mathbf{P}_{k-1}^{(l)})}$
      $\mathbf{P}_k^{(l)} = \text{Proj}(K_{\text{proj}}, \mathbf{a}^k, \mathbf{a}^{k+1})$
      $\tilde{\mathbf{P}}_k^{(l)} = \mathbf{P}_k^{(l)} \oslash \mathbf{a}^k$
    **end for**
    Calculate $(M_k^{(l+1)})_k$ by Eq. 12 with $(\tilde{\mathbf{P}}_k^{(l)})_k$
  **end for**

---

then it is satisfied that $\frac{\partial f^\top}{\partial Y} = 2Y^\top$. For the matrix $\mathbf{P}_k$, we have

$$
\begin{aligned}
df &= tr\left(\frac{\partial f^\top}{\partial Y} d \prod_{t=1}^{K} \tilde{\mathbf{P}}_t\right) \\
&= tr\left(\prod_{t_3=k+1}^{K} \tilde{\mathbf{P}}_{t_3} \frac{\partial f^\top}{\partial Y} \prod_{t_1=1}^{k} \tilde{\mathbf{P}}_{t_1} \text{Diag}\left(\frac{1}{\mathbf{a}^k}\right) d\mathbf{P}_k\right) \\
&= tr\left(2 \prod_{t_3=k+1}^{K} \tilde{\mathbf{P}}_{t_3} \left(\prod_{t_2=1}^{K} \tilde{\mathbf{P}}_{t_2} - \mathbf{I}\right)^\top \prod_{t_1=1}^{k} \tilde{\mathbf{P}}_{t_1} \text{Diag}\left(\frac{1}{\mathbf{a}^k}\right) d\mathbf{P}_k\right) \\
&= tr\left(\left(2\text{Diag}\left(\frac{1}{\mathbf{a}^k}\right)\left(\prod_{t_1=1}^{k} \tilde{\mathbf{P}}_{t_1}\right)^\top \left(\prod_{t_2=1}^{K} \tilde{\mathbf{P}}_{t_2} - \mathbf{I}\right)\left(\prod_{t_3=k+1}^{K} \tilde{\mathbf{P}}_{t_3}\right)^\top\right)^\top d\mathbf{P}_k\right)
\end{aligned}
\tag{26}
$$

thus we have

$$
M_k = 2\text{Diag}\left(\frac{1}{\mathbf{a}^k}\right)\left(\prod_{t_1=1}^{k-1} \tilde{\mathbf{P}}_{t_1}\right)^\top \left(\prod_{t_2=1}^{K} \tilde{\mathbf{P}}_{t_2} - \mathbf{I}\right)\left(\prod_{t_3=k+1}^{K} \tilde{\mathbf{P}}_{t_3}\right)^\top.
\tag{27}
$$

According to Eq. 23, we have

$$
\mathbf{P}_k = \text{Diag}(e^{\mathbf{f}_k/\epsilon}) e^{(-\mathbf{C}_k + \delta' M_k)/\epsilon} \text{Diag}(e^{\mathbf{g}_k/\epsilon})
\tag{28}
$$

and the iterative Sinkhorn algorithm can be used with the constraints given in Eq. 21.

## C  MORE DETAILS IN TSP-COT

### C.1  LAGRANGIAN METHOD FOR REGULARIZED TSP-COT

For the minimization of regularized TSP-COT, we have

$$\min_{\mathbf{P}} \mathcal{L}_2 = \langle \mathbf{C}, \mathbf{P} \rangle - \epsilon H(\mathbf{P}) + \sum_k \delta'_k ||\mathbf{P}^k - \mathbf{I}||_F^2 \tag{29}$$

$$\text{s.t.} \quad \mathbf{P}\mathbf{1}_K = \mathbf{1}_K, \ \mathbf{P}^\top \mathbf{1}_K = \mathbf{1}_K.$$

Lagrangian methods are used to solve it here. Introducing the duals $(\mathbf{f}, \mathbf{g})$ to the constraints $\mathbf{P}\mathbf{1}_K = \mathbf{1}_K, \mathbf{P}^\top \mathbf{1}_K = \mathbf{1}_K$, we can get the Lagrangian function

$$\mathcal{L} = \mathcal{L}_2 - \langle \mathbf{f}, \mathbf{P}\mathbf{1}_K - \mathbf{1}_K \rangle - \langle \mathbf{g}, \mathbf{P}^\top \mathbf{1}_K - \mathbf{1}_K \rangle. \tag{30}$$

Then we can compute the partial derivative of $\mathcal{L}$ with respect to $\mathbf{P}$ as

$$\frac{\partial \mathcal{L}}{\partial \mathbf{P}} = \mathbf{C} + \epsilon \log \mathbf{P} - \mathbf{f}\mathbf{1}^\top - \mathbf{1}\mathbf{g}^\top + M = 0, \tag{31}$$

where $M$ is specified as

$$M = \sum_k \delta'_k \frac{\partial ||\mathbf{P}^k - \mathbf{I}||_F^2}{\partial \mathbf{P}}$$

$$= \sum_{k=1}^K \sum_{t=0}^{k-1} 2\delta'_k (\mathbf{P}^t)^\top (\mathbf{P}^k - \mathbf{I})(\mathbf{P}^{k-1-t})^\top. \tag{32}$$

To prove that, we define $f_k = tr(Y^\top Y) = ||\mathbf{P}^k - \mathbf{I}||_F^2$ where $Y = \mathbf{P}^k - \mathbf{I}$, then

$$df_k = tr\left(2Y^\top dY\right) = tr(\sum_{t=0}^{k-1} \frac{\partial f_k^\top}{\partial Y}(\mathbf{P}^t d\mathbf{P}\mathbf{P}^{k-1-t}))$$

$$= tr(\sum_{t=0}^{k-1} \mathbf{P}^{k-1-t} \frac{\partial f_k^\top}{\partial Y} \mathbf{P}^t dP) \tag{33}$$

$$= tr\left(\left(2\sum_{t=0}^{k-1}(\mathbf{P}^t)^\top(\mathbf{P}^k - \mathbf{I})(\mathbf{P}^{k-1-t})^\top\right)^\top d\mathbf{P}\right),$$

Thus we have

$$\frac{\partial f_k}{\partial P} = \sum_{t=0}^{k-1} 2(\mathbf{P}^t)^\top(\mathbf{P}^k - \mathbf{I})(\mathbf{P}^{k-1-t})^\top. \tag{34}$$

So we can get that

$$M = \sum_k \delta'_k \frac{\partial f_k}{\partial \mathbf{P}} = \sum_{k=1}^K \sum_{t=0}^{k-1} 2\delta'_k (\mathbf{P}^t)^\top(\mathbf{P}^k - \mathbf{I})(\mathbf{P}^{k-1-t})^\top. \tag{35}$$

According to 31, we can the solution form

$$\mathbf{P} = \text{Diag}(e^{\mathbf{f}/\epsilon})e^{-(\mathbf{C}+M)/\epsilon}\text{Diag}(e^{\mathbf{g}/\epsilon}). \tag{36}$$

Thus the iterative Sinkhorn algorithm can be applied for calculation.

### C.2  GREEDY METHOD FOR SEARCHING WITH PROBABILITY MATRIX

With a known probability matrix calculated by Sinkhorn or Algorithm 3, we can apply the Algorithm 4 to get the TSP path.

---

**Algorithm 3:** Probability Matrix Calculation for Regularized TSP-COT.

---

**Input:** Cost Matrix $\mathbf{C}$ and iteration number $L$
**Output:** the coupling $\mathbf{P}^{(L)}$
  Initialize $M^{(0)} = \mathbf{0}_{K \times K}$
  **for** $l = 0, 1, \ldots, L$ **do**
    $\mathbf{P}^{(l)} = \text{Sinkhorn}(\mathbf{C} + M^{(l)})$
    Calculate $M^{(l+1)}$ by Eq. 16 with $\mathbf{P} = \mathbf{P}^{(l)}$
  **end for**

---

---

**Algorithm 4:** Greedy search using probability matrix to get the TSP path

---

**Input:** the coupling $\mathbf{P}$
**Output:** the path $tour$
  Initialize $tour = []$
  $i, j = \text{where}(\mathbf{P} == \mathbf{P}.max())$
  $\mathbf{P}[i,:] = \mathbf{P}[:,j] = \mathbf{P}[:,i] = 0$
  $k = j$
  $tour.append(i)$
  $tour.append(j)$
  **for** $m = 1, \ldots, n - 2$ **do**
    $i, j = k, \text{where}(\mathbf{P}[\mathbf{k},:] == \mathbf{P}[\mathbf{k},:].max())$
    $\mathbf{P}[i,:] = \mathbf{P}[:,j] = 0$
    $k = j$
    $tour.append(j)$
  **end for**

---

## D   EXPERIMENTS ON MODEL FUSION

**COT for Multi-model fusion.** Following (Singh & Jaggi, 2020) that applies the OT for model fusion task, we apply our RCOT-Sinkhorn algorithm instead of the previous pairwise Sinkhorn algorithm in (Singh & Jaggi, 2020) for multi-model fusion. Without loss of generality, here we consider fusing three models. Assume $\mathbf{W}_k^{(l,l-1)}$ is the weight matrix for model $k$ ($k = 1, 2, 3$) between layer $l$ and $l - 1$, and $\widehat{\mathbf{W}}_k^{(l,l-1)}$ ($k = 2, 3$) is the modified weights with alignments $\tilde{\mathbf{P}}_1^{l-1}, \tilde{\mathbf{P}}_3^{l-1}$ before layer $l$:

$$\widehat{\mathbf{W}}_2^{(l,l-1)} = \mathbf{W}_2^{(l,l-1)}(\tilde{\mathbf{P}}_1^{(l-1)})^\top, \quad \widehat{\mathbf{W}}_3^{(l,l-1)} = \mathbf{W}_3^{(l,l-1)}\tilde{\mathbf{P}}_3^{(l-1)}. \tag{37}$$

Then we can get the weight alignments for $\widetilde{\mathbf{W}}_2^{(l,l-1)}$ and $\widetilde{\mathbf{W}}_3^{(l,l-1)}$ to $\mathbf{W}_1^{(l,l-1)}$ by

$$\widetilde{\mathbf{W}}_2^{(l,l-1)} = \tilde{\mathbf{P}}_1^l \widehat{\mathbf{W}}_2^{(l,l-1)} \quad , \widetilde{\mathbf{W}}_3^{(l,l-1)} = (\tilde{\mathbf{P}}_3^l)^\top \widehat{\mathbf{W}}_3^{(l,l-1)}, \tag{38}$$

where $\tilde{\mathbf{P}}_1^l$ is the alignment from model 1 to model 2 and $\tilde{\mathbf{P}}_3^l$ is the alignment from model 3 to model 1 for layer $l$ calculated by RCOT-Sinkhorn. Finally, we get the parameter matrix of the fused model:

$$\mathbf{W}_\mathcal{F}^{(l,l-1)} = \frac{1}{3}\left(\mathbf{W}_1^{(l,l-1)} + \widetilde{\mathbf{W}}_2^{(l,l-1)} + \widetilde{\mathbf{W}}_3^{(l,l-1)}\right). \tag{39}$$

Initializing $l = 2$ and updating $\widehat{\mathbf{W}}_k^{(l,l-1)}$ and $\widetilde{\mathbf{W}}_k^{(l,l-1)}$ (k=2,3) by varying $l$, we can get the fused model's parameter matrices $\{\mathbf{W}_\mathcal{F}^{(l,l-1)}\}$ for predictions.

Besides, previous fusion methods are mostly based on different initialization or networks with the same task and loss. Here, we attempt to fuse models with different training methods. We combine models trained using standard training, Mixup (Zhang et al., 2017), and adversarial training (Shafahi et al., 2019), and examine the differences in their clean accuracy and robust accuracy results. The results are given in Tab. 3.

## E   GAP-GUIDED ADAPTIVE PREDICTIONS BY CALCULATING ADAPTIVE VALUE

The $\lambda$ in the algorithm can be replaced with the parameters that need to be adjusted (such as $\delta'$ and $\epsilon$).

Table 3: **Fusing standard training, mixup training and adversarial models on VGG11, along with the Top-1 accuracy of finetuning the fused models on CIFAR10.** The PGD (Madry et al., 2017) model trained here and the attack employed for obtaining robust accuracy both utilize $l_2$-PGD (PGD attack bounded with $l_2$ norm), with the perturbation size $8.0/255$, which is also used for robust accuracy.

| ACC | $M_A$ (ST) | $M_B$ (Mixup) | $M_C$ (PGD) | Model Fusion without Fine-tuning | | | | Standard Fine-tuning | | | Mixup Fine-tuning | | | PGD Fine-tuning | | |
|---|---|---|---|---|---|---|---|---|---|---|---|---|---|---|---|---|
| | | | | PREDICTION | VANILLA | OTfusion | Ours | VANILLA | OTfusion | Ours | VANILLA | OTfusion | Ours | VANILLA | OTfusion | Ours |
| Clean | 90.49 | 92.30 | 89.24 | 91.54 | 42.48 | 90.21 | 89.88 | 89.71 | 89.88 | **89.96** | 89.67 | 89.66 | **90.40** | 89.88 | 89.86 | **90.62** |
| Robust | 30.76 | 49.84 | 73.73 | / | 29.66 | 30.57 | 30.68 | 29.95 | 30.63 | **31.84** | 49.04 | 51.60 | **52.36** | 50.15 | 50.75 | **53.07** |

| Original | After $\mathbf{P}_1$ | After $\mathbf{P}_2$ | After $\mathbf{P}_3$ | After $\mathbf{P}_4$ |
|---|---|---|---|---|

Figure 7: Example point matching results after applying $\mathbf{P}_1$, $\mathbf{P}_2$, $\mathbf{P}_3$ and $\mathbf{P}_4$ to the original point set when switching the second and third set. Compared to Fig. 4, the image order is switched.

---

**Algorithm 5:** gap-guided adaptive predictions by calculating adaptive value

---

**Input:** the performance gap between COT (with $\lambda = x$) and Concorde: $gap(x)$
**Output:** the adaptive value $x^*$

1 Initialize: $x_1 = 0.001, x_2 = 0.01, \delta = 0.0001 and \epsilon = 0.0001$
2 calculate d$gap_1 = gap(x_1 + \delta) - gap(x_1)$
3 calculate d$gap_2 = gap(x_2 + \delta) - gap(x_2)$
    /* Here we can check that d$gap_1 \cdot$d$gap_2 < 0$                         */
4 $\bar{x} = (x_1 + x_2)/2$
5 **while** $x_2 - x_1 > \epsilon$ **do**
6      calculate d$gap_m = gap(\bar{x} + \delta) - gap(\bar{x})$
7      **if** $dgam_1 \cdot dgap_m < 0$ **then**
8          $x_2 = \bar{x}$
9          d$gap_2 =$d$gap_m$
10      **else if** $dgam_2 \cdot dgap_m < 0$ **then**
11          $x_1 = \bar{x}$
12          d$gap_1 =$d$gap_m$
13      **else**
14          break
15      **end**
16      $\bar{x} = (x_1 + x_2)/2$
17 **end**
18 the adaptive variable $x^* = \bar{x}$

---

# F DEFINITION OF CONVERGENCE ORDER AND CONVERGENCE LINE

Suppose $\{p_n\}_{n=0}^{\infty}$ is a sequence that converges to p, and for each n, $p_n \neq p$. If there exists positive constant $\lambda$ and $\alpha$ such that

$$\lim_{n \to \infty} \frac{|p_{n+1} - p_n|}{|p_n - p|^{\alpha}} = \lambda$$

then the sequence $\{p_n\}_{n=0}^{\infty}$ converges to p at order $\alpha$, and the corresponding constant $\lambda$ is referred to as the asymptotic error constant.

To validate the convergence order of $\{p_n\}_{n=0}^{\infty}$, we plot the $\alpha$-order convergence line $\log(|p_n - p_{n-1}|^{\alpha})$ and $\log(|p_{n+1} - p_n|)$ on the same graph. Then $\{p_n\}_{n=0}^{\infty}$ converging at order $\alpha$ is equivalent to the $\alpha$-order convergence line being parallel to the line of $\log(|p_{n+1} - p_n|)$.

## G    MORE EXPERIMENTAL RESULTS

Table 4:    **The results for the case of three measures (K=3 ) when switching the order on Willow_3GM with IPCA-GM as backbone.**

| Willow_3GM with IPCA-GM | ACC | CACC | CR |
|---|---|---|---|
| RCOT-Sinkhorn : A→B, B→C, C→A | 0.9506 | 0.9031 | 0.9500 |
| RCOT-Sinkhorn : A→C, C→B, B→A | 0.9459 | 0.8974 | 0.9477 |

Table 5:    **The results for the case of four measures (K=4) when switching the second and third set order on Willow_4GM with IPCA-GM as backbone.**

| Willow_4GM with IPCA-GM | ACC | CACC | CR |
|---|---|---|---|
| RCOT-Sinkhorn without switching | 0.9453 | 0.8592 | 0.9240 |
| RCOT-Sinkhorn with switching the order | 0.9460 | 0.8557 | 0.9222 |

Table 6:    **Ablation study for visual matching experiments by varying $\delta$ and $\epsilon$.**

| $\delta$ | $\epsilon$ | ACC | CACC | CR |
|---|---|---|---|---|
| 0.001 | 1e-9 | 0.9412 | 0.8767 | 0.9158 |
| 0.001 | 1e-10 | 0.9412 | 0.8767 | 0.9158 |
| 0.01 | 1e-9 | 0.9442 | 0.8967 | 0.9475 |
| 0.001 | 1e-11 | 0.9412 | 0.8767 | 0.9158 |
| 0.01 | 1e-10 | 0.9442 | 0.8967 | 0.9475 |
| 0.01 | 1e-11 | 0.9442 | 0.8967 | 0.9475 |
| 0.1 | 1e-9 | 0.9382 | 0.9087 | 0.9951 |
| 0.1 | 1e-10 | 0.9382 | 0.9087 | 0.9951 |
| 0.1 | 1e-11 | 0.9382 | 0.9087 | 0.9951 |

## H    FURTHER DISCUSSIONS

### H.1    THE EXISTENCE OF A SOLUTION TO MONGE FORMULATION IN EQ. 6

A solution to the Monge formulation in Eq. 6 does exist. We can consider a feasible solution as follows: assume that $\{t_1, t_2, \ldots, t_{K-1}\}$ are the solutions of the original MMOT problem. Then, given $x \in \mathcal{X}_1$ and $y = T_{K-1}T_{K-2}\cdots T_1(x)$, we can set $x = T_K(y)$. In this way, it satisfies the conditions and thus is a feasible solution.

### H.2    THE CONNECTION BETWEEN MMOT AND COT

Take the case when $K = 3$ as an example. In MMOT, the formulation is given by $\min_X \sum_{ijk} C_{ijk} X_{ijk}$, s.t. $\sum_{ij} X_{ijk} = 1, \sum_{ik} X_{ijk} = 1, \sum_{jk} X_{ijk} = 1$.

While in COT, the formulation is given by $\min \sum_k \langle D_k, P_k \rangle$, s.t. $\forall k, P_k \mathbf{1}_K = \mathbf{1}_K, P_k^T \mathbf{1}_K = \mathbf{1}_K$, and $P_1 P_2 P_3 = \mathbf{I}$.

Let $C_{ijk} = D_{1ij} + D_{2jk} + D_{3ik}$. Then, the objective function of MMOT can be written as $\sum_{ijk}(D_{1ij} + D_{2jk} + D_{3ik}) X_{ijk}$, which, through algebraic manipulation, equals $\sum_{ij} D_{1ij} \sum_k X_{ijk} + \sum_{jk} D_{2jk} \sum_i X_{jk} + \sum_{ik} D_{3ik} \sum_j X_{ik}$.

By introducing the notations $\sum_k X_{ijk} = P_{1ij}$, $\sum_i X_{ijk} = P_{2jk}$, and $\sum_j X_{ijk} = P_{3ik}$, the objective function is transformed into $\min \sum_k \langle D_k, P_k \rangle$, which clearly exhibits the COT form.

When $K > 3$, the MMOT can also be transformed into COT by using a similar method. This shows that, under specific definitions and transformations of the cost matrices and variables, MMOT can be related to COT in a structured way. Although MMOT deals with a higher-dimensional tensor involving all distributions leading to high computational costs, our COT, through this demonstrated connection, can leverage certain aspects of MMOT's framework while maintaining its own computational efficiency and applicability in the targeted problems.

### H.3 DEPENDENCE ON THE ORDER OF MEASURES $\alpha$ IN COT FORMULATIONS

It is obvious that the order of measures has no impact when $K < 3$. For $K = 3, 4$, as shown in 4.3, the results before and after switching the order of measures are almost the same, demonstrating that the order of measures has little impact on the results for $K = 3, 4$.

It is important to note that in both the datasets we utilized and the majority of practical applications within our research domain, the value of $K$ typically does not exceed 4. This practical constraint implies that the scenarios we are primarily concerned with are well-covered by our existing experimental setup.

### H.4 THE DEGREE OF SATISFACTION OF CYCLE-CONSISTENCY

We use Consistent Rate (**CR**) defined in Eq. 18 to assess the degree of satisfaction of cycle-consistency in Table 1. The closer **CR** is to $100\%$, the better cycle-consistency is achieved.

