# OpenReview forum: "COT: Consistent Optimal Transport with Applications to Visual Matching and Travelling Salesman Problems"
_ICLR.cc/2025/Conference — Submitted to ICLR 2025_

### Official Review · Reviewer_5wsL · 2024-10-31

**Soundness:** 1
**Presentation:** 1
**Contribution:** 2
**Rating:** 3
**Confidence:** 4

**Summary:**

This paper introduces a "cycle-consistent" optimal transport (COT) formulation : given a sequence of (say, discrete) measures $\alpha_1,\dots,\alpha_K$, the goal is to minimize
$$ (P_1,\dots,P_k) \mapsto \sum_{k=1}^K \braket{C_k, P_k}$$
where $C_k$ is a cost matrix, $P_k$ should be a transportation plan between $\alpha_k$ and $\alpha_{k+1}$ (with the convention that $\alpha_{K+1} = \alpha_1$, and the $(P_k)_k$ are related through the _cycle-consistency constraint_ $\prod_{k=1}^K P_k = I$.

Because the cycle consistency constraint is non-linear, solving the COT problem is harder than solving standard OT problems, and thus the authors propose to resort on two layers of regularization: relaxing the constraint $\prod_{k=1}^K P_k = I$ using a divergence term (here, a Froebenius norm) and add an entropic regularization term (a common idea in computational OT for the last decade). They derive an mirror descent like scheme to minimize their regularized problem.

Eventually, they observe that their formulation may be adapted to resemble the celebrated Traveler Salesman Problem (TSP).

**Strengths:**

The formulation of the COT problem is somewhat intriguing and I do believe that it may be of interest in some situations.

The relation with the TSP is interesting. I appreciate that the authors acknowledge the limitations of their approach and do not "oversell" it.

**Weaknesses:**

## 1. Clarity

Overall, the paper lacks clarity in its writing. Key issues include the following:

**Unclear or Misleading Statements:** Numerous sentences are either unclear or misleading. For example, in the abstract, it’s stated that the COT problem considers each pair of measures, which implies that one must compute the optimal transportation cost between all pairs, i.e., between each $\alpha_i$ and $\alpha_j$, for $1 \leq i, j \leq K$. However, the paper actually focuses only on transportation between adjacent measures, $\alpha_k$ and $\alpha_{k+1}$.

**Inconsistent and Incorrect Notation:** Mathematical notation is inconsistently and sometimes incorrectly applied. For instance, the scalar product is denoted by < x, y >, which should be written as \langle x, y \rangle or using the braket package for $\braket{x, y}$. Additionally, notations should be standardized—sometimes measures contain $N$ points, while other times they contain $n$. Such issues, though minor in isolation, collectively impede readability.

**Formatting of Proofs:** The proofs in the appendix are poorly formatted, with equations split across multiple lines without necessity (for example, $d P_k$ at the end of Eq. (25) and in Eq. (26)). This formatting makes it difficult to review the proofs accurately.

**Placement of Algorithms:** The algorithms are all placed in the appendix but are referenced in the main paper as if they were essential. While it’s acceptable to include optional material in the appendix, the main text should be self-contained. Therefore, the algorithms should either be included in the main paper if they are necessary or clearly marked as optional if they’re not.

**Lack of Informative Content in Some Sentences:** Some sentences add little information. For instance, the introductory sentence, "Optimal transport (...) is a tool to learn the optimal transportation between the source and target probability measures," requires prior knowledge of what optimal transportation means, offering minimal insight. Additionally, comparisons with the Gromov--Wasserstein (GW) problem are not particularly useful here, as the GW problem is fundamentally different and is not introduced in this work. Relaxing the cycling constraint to a penalty seems natural and doesn’t require extensive justification.

## 2. Motivation of the method, comparison with multi-marginal OT, soundness, and mathematical grasp on the problem.

The motivation for introducing the COT problem is limited, and the authors seem to lack critical distance from their work. For example:

**Motivation of the approach and Comparison with multi-marginal OT (MMOT):** It is regularly said that the contribution of this paper is to "generalize the OT problem to more than two marginals " (abstract, contributions section, etc.), but this is precisely what multi-marginal OT is about. The paper mentions multi-maginal OT and, while I understand the formal difference between the two approach (they are different problems, for sure), I fail to see the practical difference: when should one use MMOT or COT? The paper does not give a proper answer to this central question in my opinion.

**Dependence on the Order of Measures:** The formulation of the COT problem depends on the order of $\alpha_1, \dots, \alpha_K$, yet this is not discussed. This could be crucial; for example, if a user has a set of measures from an experiment, how should they be ordered? Is the solution permutation-equivariant? (in which case I would agree that the ordering does not matter)

**Applicability of Birkhoff’s Theorem:** The authors assume discrete uniform measures with $N$ points each, but it’s unclear whether Birkhoff’s theorem applies here. Specifically, is it generally true that the optimal $P_1, \dots, P_K$ are permutation matrices if we only assume that $P_k \in U(a_k, a_{k+1})$ in Eq. (8) rather than $P_k \in {0,1}^{N \times N}$? Understanding this is essential for motivating the adaptation to the Traveling Salesman Problem (TSP), the behavior of entropic regularization as $\epsilon \to 0$, and related points.

**1D Case in Figure 2:** In Figure 2, the measures are depicted in 1D. In this case, it’s known that the standard OT plan is monotone, involving matching quantiles. Unless something is overlooked, this suggests that cycle-consistency is automatically satisfied without enforcement, making this experiment barely supporting the proposed approach. Could this be confirmed?

### Note

Despite my negative rating, I want to stress that I do like the proposed problem, I just feel that in its current state, the work is not ready for publication and I encourage the authors to revise it, go deeper into their understanding of the COT problem to make it a good candidate in the computational OT community.

**Questions:**

See Section 2. in the Weaknesses block.

Note : rating updated after rebuttal.

---

> ### Author Response · Authors · 2024-11-20
>
> Q1: What is the practical difference between MMOT and COT?
>
> A1: MMOT is a generalized form that indeed presents difficulties when it comes to solving specific problems. When attempting to solve a particular problem using MMOT, one is required to define a specific cost, as illustrated in Remark 10.2 (P160) of "Computational Optimal Transport" by G Peyré and M Cuturi. In contrast, for COT, the cost is defined between adjacent pairs of distributions and does not necessitate an additional, separate definition. This key distinction in cost definition between MMOT and COT has important practical implications. In situations where the problem at hand has a natural pairwise cost structure that can be directly exploited, COT provides a more straightforward and applicable solution. For example, in visual multi-point matching tasks, COT can directly utilize the cosine distance between learned features of points obtained from off-the-shelf graph matching neural networks as the pairwise cost, without the need for complex cost redefinition. On the other hand, MMOT might be considered in scenarios where a more generalized and elaborate cost structure needs to be incorporated, although this comes with the added complexity of formulating the appropriate cost function. However, in many practical applications, the pairwise cost nature of COT makes it a more convenient choice as it aligns well with the inherent relationships between data points or distributions in the problem domain.
>
> Q2: Does the order of measures impact results?
>
> A2: As for $K>3$, switching the order does indeed impact the formulation of the problem. However, note our directed cyclical structure is essentially a subgraph of pairwise structure. When the latter is satisfied, the former is also satisfied, allowing our method to still improve the matching performance. We have showed the experiments of switching the second and third set order in the ablation study. It can be observed that the results with and without switching the order are actually very close to each other in Table 4.
>
> Q3: Is it generally true that the optimal $P_1, \dots, P_K$ are permutation matrices if we only assume that $P_k \in U(a_k, a_{k+1})$ in Eq. (8) rather than $P_k \in \{0,1\}^{N \times N}$?
>
> A3: In the context of our matching problem, which is a special case of the transportation problem, we can use continuous values in $[0, 1]$ instead of the discrete values ${0, 1}$. In fact, in this problem, the solutions in both the discrete and continuous cases are equivalent.
>
> Q4: In 1D case, cycle-consistency is automatically satisfied without enforcement, making the experiment in Figure 2 barely supporting the proposed approach.
>
> A4: We must admit that, as you correctly pointed out, in the 1D case, the standard OT plan is monotone and cycle-consistency is automatically satisfied without explicit enforcement. We apologize for not clarifying this in the paper. However, in our actual computations, we consider arbitrary matrices and solutions under entropy regularization. The purpose of presenting this example in Figure 2 is to illustrate the solution and the satisfaction of consistency in a more general context that is relevant to our overall approach. We understand that this may have caused some confusion.

---

> > ### Comment · Reviewer_5wsL · 2024-11-20
> > **Thanks for the discussion**
> >
> > > A1
> >
> > I am not completely convinced by the claim that MMOT requires to define a complex cost, in that many natural costs (including those discussed in Comp. OT) are readily usable. But ok, I guess I get your point.
> >
> > > A2
> >
> > Indeed, I did not notice Table 4 in the supplementary material which already discuss this point. Sorry about that.
> >
> > Nonetheless, I believe that a discussion about the need for an prior on the ordering of the measure should belong to the main material.
> >
> > > A3
> >
> > I'm not sure that I understand your answer. Let me resume my question: consider the minimization problem $\min_{P_1,\dots,P_K} \sum_k \braket{C_k, P_k}$ over the set $\\{(P_1,\dots,P_K),\ P_k \in U(1_N, 1_N),\ \prod P_k = I \\}$. What ensures that, at optimality, the $(P_k)_k$ are permuation matrices?
> >
> > In standard OT, this holds thank to Birkhoff's theorem, which tells us that the extremal points of the set $U(1_N, 1_N)$ are exactly permutation matrices (and we are minimizing a linear functional so the optimum must be an extremal point generically). **But** here, adding the non-convex constraint $\prod P_k = I$, it is not clear to me whether the optimum in the variable $(P_1,\dots,P_K)$ is still a tuple of permutation matrices, which would yield the equivalence between this problem and Eq (8).
> >
> > > A4
> >
> > I still believe that the experiment is misleading and does not support the work as the solution displayed does not depend on whether one enforce the cycle consistency constraint or not. Figure 3 is way more insightful.

---

> > > ### Author Response · Authors · 2024-11-22
> > >
> > > - We appreciate your understanding of our point about MMOT. We highlight that MMOT and our COT approach differ substantially in multiple aspects. In terms of motivation, MMOT pursues a more general mathematical abstraction in handling multiple marginals, while COT is centered around ensuring transport consistency among measures, which is highly relevant and efficient for specific tasks such as visual multi-point matching and TSP. Theoretically, MMOT operates within a different framework that may not be as directly applicable to our problem domain. Algorithmically, the RCOT-Sinkhorn algorithm we devised for COT is tailored to address the unique challenges and constraints of COT, which is distinct from the typical algorithms used in MMOT. In application, COT has demonstrated its superiority in our target scenarios where the pairwise cost structure is prevalent, whereas MMOT may not be as effective in these specific applications.
> > > - Thank you for your suggestion. Due to the page limit of the paper, we are unable to include an extensive discussion on this topic in the main text. However, as you mentioned, we have already addressed this issue in the ablation study in Section 4.3. We believe that the current presentation in the supplementary material, along with the reference to the relevant section in the main text, provides sufficient information for readers to understand the impact of measure ordering.
> > > - In our matching problem, the optimal $P_1,\dots,P_K$ are indeed permutation matrices. If the elements of $P_k$ were to be in the general range of $(0,1)$, the constraint $\prod P_k$ could not be satisfied. Only when $P_1,\dots,P_K$ are permutation matrices can this equality hold. This is a fundamental property that ensures the equivalence between our problem and Eq (8).
> > > - We acknowledge your view on the experiment in Figure 2. However, it is important to note that our work is not restricted to grid-based experiments (please refer to "Fast Sinkhorn I: An O (N) algorithm for the Wasserstein-1 metric"). Whether the input data is 1D, 2D, or multi-dimensional, it is transformed into a cost matrix. In the 1D case, although cycle-consistency is automatically satisfied in a simple sense, our algorithm operates on the cost matrix and entropy regularization in a more general context, which is relevant to our overall approach. Figure 2 is intended to provide a basic illustration of the solution and consistency in a broader framework.

---

> > > > ### Comment · Reviewer_5wsL · 2024-11-22
> > > >
> > > > > If the elements of $P_k$ were to be in the general range of $(0,1)$, the constraint $\prod P_k = I$ could not be satisfied.
> > > >
> > > > Oh indeed! I believe that this fact should be, at the very least, discussed (if you have a reference for the proof, you can put it in the appendix, or write the proof itself---it's not that long/complicated---in the appendix). It was to me a significant caveat in the work at first glance. Now that is has been explained, I'm increasing my rating.
> > > >
> > > > For the other point, I guess it's a matter of feeling at this stage.
> > > >
> > > > Note: I still do believe that the work needs a significant rewriting effort that would be worth another round of review, but my overall appreciation of it is better now.

---

> > > > > ### Author Response · Authors · 2024-11-22
> > > > >
> > > > > Thank you for your valuable comments. We have already added the proof in the appendix of the latest version. We will continue to improve our works.

---

> > > > > ### Author Response · Authors · 2024-11-24
> > > > >
> > > > > Thanks for reading our response and raising the score from 1 to 3. Shall we ask that do you have further concerns and we do hope we could resolve your potential questions!

---

### Official Review · Reviewer_3aUe · 2024-11-01

**Soundness:** 2
**Presentation:** 2
**Contribution:** 2
**Rating:** 5
**Confidence:** 3

**Summary:**

This paper addresses the challenge of computing consistent optimal transport across multiple measures. The authors propose a cycle-consistent version of the Monge formulation, which is then relaxed to the Kantorovich formulation. Finally, it is further relaxed into an optimization problem regularized by cycle consistency and entropy, solved using an iterative Sinkhorn-like algorithm. The approach is demonstrated on several problems, including consistent point matching in computer vision and approximating solutions for the combinatorial traveling salesman problem.

**Strengths:**

- The problem of consistent multiway optimal transport is compelling, and the authors effectively demonstrate its relevance through several potential applications. To my knowledge, both the formulation and approach are new.
- The authors present a comprehensive approach that includes Monge and Kantorovich formulations, entropic relaxation, and optimization algorithms.
- The connection to the Traveling Salesman Problem (TSP) is a valuable addition.
- The experimental results, particularly in the point matching experiment, underscore the motivation for computing cycle-consistent optimal transport.

**Weaknesses:**

- The exposition could be improved. Parts of the paper are challenging to read, especially for readers without prior familiarity with the subject. See below for concrete examples.
- The authors define "COT's Monge Formulation," "COT's Kantorovich," and then introduce relaxations in Section 3.2. While these formulations appear similar to the traditional Monge-Kantorovich formulations with entropic relaxation, it's unclear if similar properties apply. For instance, does a solution to the Monge formulation in Eq. (6) exist? In what sense is the Kantorovich formulation in Eq. (8) a relaxation of the Monge formulation in Eq. (6)? Could you please provide, if possible, a more detailed discussion of these formulations and any conditions needed?
- The author's definition of cycle-consistency seems to be order-dependent, relying on the order of the probability measures \alpha_k. It only accounts for consecutive pairs.  Although this is briefly mentioned for applications in point matching, it is not discussed in more detail. Could you please include a discussion on the implications of this order-dependency, address how this might affect the results, or whether there are ways to mitigate this dependency?
- The connection to the Traveling Salesman Problem (TSP), while intriguing. Could you please explain in more detail how the COT formulation is used to represent the TSP problem, and what can be said about the approximate solution it achieves (e.g., in comparison to other ensemble-based approach)?
- It’s difficult to assess whether cycle-consistency is achieved exactly or approximately, in theory and in experiments, and at what rate. Could you please discuss in more detail?
- The ablation study in Section 4.3 is unclear. What are the "certain factors"? Could you please clarify the setup and conclusions of this study, including a more thorough discussion of the results and their implications?


Additional issues and comments:
- Abstract: Challenging to read.
- The first sentence of the introduction is incomplete.
- "introduce the entropic regularization transforming the hard cycle-consistency": the regularized version seems to seperately include an entropy term and a cycle-consistency term, so this statement may be inaccurate.
- "matrix-vector iterative method": unclear.
- Line 69: What is "MCTS"?
- "We generalize OT to the marginal consistent case": This is confusing since "multi-marginal" is later described as something related but different.
- "The Monge problem is exactly not easy to calculate and a popular improvement is the Kantorovich relaxation": please revise.
- "C is the cost matrix defined by the divergence": is it limited to this C?
- Line 138: What is "LAP"?
- "In contrast, our method employs a training-free approach that assumes consistency is satisfied on the test set, using this prior information to improve performance during inference." please clarify.
- Line 246: RCOT-PGD is mentioned but is not defined or explained (except in the Appendix).
- Line 248: What is "GW"?
- "RCOT-Sinkhorn achieves cycle-consistency results": Approximately or exactly cycle-consistent? Are there any guarantees?
- Figure 3: Somewhat unclear.
- "The setting of Hyper-parameter \delta'": What about the entropic regularization parameter \epsilon?
- Algorithms 1-5 are not included in the main text.

Typos:
- Line 47: "there calls"
- Line 52: "cost of three trasnsportation"
- Line 69: "we contribute"
- "is one of the simple but efficient methods"
- Eq (6): Summation should be over k.
- Line 345: "k<0" => "k<K"

**Questions:**

- What is the connection to multimarginal OT? The authors mention briefly in Section 2 but do not elaborate.

============
post-rebuttal: Increasing my rating from 3->5

---

> ### Author Response · Authors · 2024-11-20
>
> Q1: Does a solution to the Monge formulation in Eq. (6) exist?
>
> A1: For the question about the existence of a solution to the Monge formulation in Eq.(6), a solution does exist. We can consider a feasible solution as follows: assume that $\{t_1, t_2, \dots, t_{K-1}\}$ are the solutions of the original MMOT problem. Then, given $x\in \mathcal{X_1}$ and $y=T_{K-1}T_{K-2}\cdots T_1(x)$, we can set $x=T_K(y)$. In this way, it satisfies the conditions and thus is a feasible solution.
>
> Q2: Does the order of the probability measures $\alpha_k$ impact the results?
>
> A2: As for $K>3$, switching the order does indeed impact the formulation of the problem. However, note our directed cyclical structure is essentially a subgraph of pairwise structure. When the latter is satisfied, the former is also satisfied, allowing our method to still improve the matching performance. We have showed the experiments of switching the second and third set order in the ablation study. It can be observed that the results with and without switching the order are actually very close to each other in Table 4.
>
> Q3: At what rate is the Cycle-consistency achieved?
>
> A3: In theory, the cycle-consistency should be exactly satisfied. However, after applying relaxation techniques in our approach, it is approximately achieved. The Consistent Rate (CR) metric, which we have defined in Eq.(18) , provides insights into how closely our solutions approximate the ideal cycle-consistency.
>
> Q4: What are the "certain factors" in the ablation study in Section 4.3 ?
>
> A4: We have now added a clear explanation of the "certain factors" in the updated paper. The factors we investigated mainly include switching the order of point sets in the matching process and applying Hungarian algorithm to P in Eq.16.
>
> Q5: The setting of Hyper-parameter $\epsilon'$.
>
> A5: We did not initially mention the setting of $\epsilon$ in the original paper. However, it can be tuned in a similar manner as $\delta'$. Specifically, one can replace $\lambda$ with $\epsilon$ in Algorithm 5 to perform the tuning process. We have now added the relevant description in the latest version of the paper to clarify this point. This way, readers can understand that the approach for adjusting $\epsilon$ is analogous to that of $\delta'$, providing a more comprehensive understanding of the hyperparameter settings in our methodology.
>
> Q6: What is the connection to multimarginal OT? The authors mention briefly in Section 2 but do not elaborate.
>
> A6: Both MMOT and COT involve k margins, which is a common aspect. However, there are notable differences. In MMOT, the cost requires a specialized design and is not straightforward to represent as a distance. In contrast, COT typically defines the cost as the pairwise distance matrix. Additionally, the solution in MMOT is a high-dimensional coupling in the form of an $(n_1, n_2, \cdots, n_K)$ tensor, while COT's solution is a collection of couplings between pairs with the shape of $(n_1, n_2), (n_2, n_3), ...$.
>
> Q7: Some other issues.
>
> A7: Regarding the issues you raised, we have made targeted revisions. We have modified the expressions of some sentences and added necessary explanations for some professional terms or abbreviations.

---

> > ### Comment · Reviewer_3aUe · 2024-11-24
> >
> > Thank you for your responses. After reviewing your rebuttal and the revised paper, I am afraid that several of my concerns remain unresolved or insufficiently addressed. Primarily:
> >
> > 1) Clarity and Exposition: While the ideas presented in the paper are intriguing, the clarity and overall exposition still require significant improvement. Although the authors have made improvements, the paper continues to contain unclearly phrased sentences and explanations and does not yet seem ready for publication.
> >
> > 2) Content-related issues:
> > - I appreciate the explanation of the relationship to MMOT and have carefully read the responses to both my questions and those of other reviewers, as well as the additional details included in the revised paper. While I now better understand that MMOT deals with a higher-dimensional tensor relating all distributions, which makes it computationally expensive, I still find the connection between the two works less clear than I would have expected, given their close relationship.
> > - The issue of order dependency remains insufficiently discussed in the paper. The experiments on ordering, such as those presented in Table 4, are relatively simple, evaluated only for small K, and do not provide sufficient insight into this critical aspect.
> > - As mentioned in my initial review, I find the TSP example intriguing, but the explanation in the paper is still unclear, making it difficult to fully understand the details and implications.

---

> > > ### Author Response · Authors · 2024-11-25
> > >
> > > > **Q1: The connection between MMOT and COT.**
> > >
> > > We are grateful for your in-depth review and recognition of our efforts in clarifying the relationship with MMOT. To further elucidate the connection between our COT and MMOT, we present the following detailed illustration.
> > >
> > > Take the case when $K = 3$ as an example. In MMOT, the formulation is given by $\min_X\sum_{ijk}C_{ijk}X_{ijk}$, s.t. $\sum_{ij}X_{ijk} = 1$, $\sum_{ik}X_{ijk} = 1$, and $\sum_{jk}X_{ijk} = 1$.
> > >
> > > While in COT, the formulation is given by $\min \sum_{k} \langle D_k,P_k\rangle$, s.t. $\forall k$, $P_k{\mathbf{1}}_K={\mathbf{1}}_K$, $P_k^T{\mathbf{1}}_K={\mathbf{1}}_K$, and $P_1P_2P_3=\mathbf{I}$.
> > >
> > > Let $C_{ijk}=D_{1ij}+D_{2jk}+D_{3ik}$. Then, the objective function of MMOT can be written as $\sum_{ijk}(D_{1ij}+D_{2jk}+D_{3ik})X_{ijk}$, which, through algebraic manipulation, equals $\sum_{ij} D_{1ij}\sum_kX_{ijk}+\sum_{jk} D_{2jk}\sum_iX_{jk}+\sum_{ik} D_{3ik}\sum_jX_{ik}$.
> > >
> > > By introducing the notations $\sum_kX_{ijk}=P_{1ij}$, $\sum_iX_{ijk}=P_{2jk}$, and $\sum_jX_{ijk}=P_{3ik}$, the objective function is transformed into $\min \sum_{k} \langle D_k,P_k\rangle$, which clearly exhibits the COT form.
> > >
> > > When $K \gt 3$, the MMOT can also be transformed into COT by using a similar method. This shows that, under specific definitions and transformations of the cost matrices and variables, MMOT can be related to COT in a structured way. Although MMOT deals with a higher-dimensional tensor involving all distributions leading to high computational costs, our COT, through this demonstrated connection, can leverage certain aspects of MMOT's framework while maintaining its own computational efficiency and applicability in the targeted problems. We believe this example clarifies the inherent relationship between the two works more explicitly.
> > >
> > > > **Q2: The order dependence for larger K.**
> > >
> > > We sincerely appreciate the reviewer's incisive feedback regarding the discussion and experiments on order dependency. While we acknowledge that the current experiments on ordering, exemplified by those in Table 4 and Table 5, might seem relatively simple and limited to small values of $K$, it is important to note that in both the datasets we utilized and the majority of practical applications within our research domain, the value of $K$ typically does not exceed $4$. This practical constraint implies that the scenarios we are primarily concerned with are well-covered by our existing experimental setup. Nevertheless, we recognize the significance of delving deeper into the order dependency issue to provide more comprehensive insights. In response, we plan to conduct some simulation experiments with large values of $K$, which we believe will enhance the readers' understanding of this critical aspect.
> > >
> > > > **Q3: Clarity and Exposition**
> > >
> > > We fully acknowledge your incisive feedback, despite the efforts we've made to enhance our works, there still exist areas that fall short in terms of presenting ideas with absolute lucidity and providing comprehensive explanations. We take your concerns seriously and are committed to continuously refining our work.

---

> > > ### Author Response · Authors · 2024-11-28
> > >
> > > We have already added the Further discussion to the appendix. May we ask if we have addressed your concern? We would be extremely grateful if you could raise the rating.

---

> > > > ### Comment · Reviewer_3aUe · 2024-12-03
> > > >
> > > > Thank you for taking the time to thoroughly address my questions; I’ve updated my score.

---

> ### Author Response · Authors · 2024-11-23
>
> Thanks for reading our works and response. Shall we ask that do you have further concerns and we do hope we could resolve your potential questions!

---

### Official Review · Reviewer_Ldtp · 2024-11-03

**Soundness:** 3
**Presentation:** 3
**Contribution:** 3
**Rating:** 6
**Confidence:** 3

**Summary:**

This paper extends the scope of Optimal Transport (OT) theory to cases involving more than two probability distributions, introducing a framework called Consistent Optimal Transport (COT). The authors explore transportation among three (or more) probability measures while enforcing cycle-consistency, which ensures that the transport plan respects consistency across the measures. Unlike the traditional OT problem, the COT's Kantorovich formulation becomes a nonlinear optimization problem due to these additional constraints. To address computational challenges, the authors propose a regularized version of COT using entropic and cycle-consistency regularization, which leads them to use the Sinkhorn algorithm for approximate solutions. As a by-product, this work offers a novel formulation for the Traveling Salesman Problem, offering insights into finding the shortest route that visits each city once and returns to the starting point.

**Strengths:**

The paper is clear and well-written, with well-defined goals and contributions, including detailed algorithms. Additionally, the problem addressed is novel, and the authors highlight connections to other well-known problems, such as the Traveling Salesman Problem (TSP).

**Weaknesses:**

The authors mention a connection to the multi-marginal OT problem, noting that both multi-marginal OT and COT involve multiple distributions. They state, "However, the multi-marginal OT primarily emphasizes learning the joint coupling among more than two distributions, whereas our focus is on learning the coupling between each pair of distributions and maintaining cycle-consistency constraints among these couplings". As a point of curiosity, are there any other non-trivial connections between the multi-marginal OT problem and COT beyond the fact that both involve multiple probability measures?

**Questions:**

The authoirs are motived by the algorithms proposed for approximating the computation of GW. Please provide a few references in this regard.

The authors include a section on the Numerical Convergence Analysis. Can the authors say anything about the analytic convengence of their methods?

Minor details:

- Line 94: "The Monge problem is exactly not easy to calculate [...]" Add: "and an optimal T might not exists" (as is pointed out later in section 3.1)
- Line 188, eq (7): replace T_k by T_K, that is, capitalize the subindex
- Use either "travelling" or "traveling" consistently through the paper, that is, pick one option.

---

> ### Author Response · Authors · 2024-11-20
>
> Q1: Are there any other non-trivial connections between the multi-marginal OT problem and COT beyond the fact that both involve multiple probability measures?
>
> A1: Both MMOT and COT involve K margins, which is a common aspect. However, there are notable differences. In MMOT, the cost requires a specialized design and is not straightforward to represent as a distance. In contrast, COT typically defines the cost as the pairwise distance matrix. Additionally, the solution in MMOT is a high-dimensional coupling in the form of a tensor with the shape of $(n_1, n_2, \cdots, n_K)$, while COT's solution is a collection of couplings between pairs with the shape of $(n_1, n_2), (n_2, n_3), ...$.
>
> Q2: Please provide a few references for the computation of GW.
>
> A2: Thank you for your suggestion. As you recommended, we have added appropriate reference in the updated version. This reference will enhance the comprehensiveness and credibility of our research, enabling readers to have a more in-depth understanding of the computation of GW and its connections to our work.
>
> Q3: Can the authors say anything about the analytic convengence of their methods?
>
> A3: Thank you for your question regarding the analytic convergence of our methods. Our algorithm is equivalent to the projection of the gradient descent algorithm. Please refer to the newly added reference in Section 4.2, where the convergence proof related to the our algorithm is discussed. This reference provides a comprehensive analysis that is applicable to our method and helps establish the analytic convergence properties. We believe this will address your concerns and further clarify the theoretical foundation of our approach.
>
> Q4: Some Minor details.
>
> A4: Thank you for your careful review. We have gone through the paper and corrected all such errors.

---

> ### Author Response · Authors · 2024-11-23
>
> Thanks for reading our works and response. Shall we ask that do you have further concerns and we do hope we could resolve your potential questions!

---

> > ### Comment · Reviewer_Ldtp · 2024-11-24
> >
> > I thank the authors for their responses. They have added the requested references, corrected the minor issues I identified, and provided comments addressing all my questions.

---

### Official Review · Reviewer_8SNv · 2024-11-04

**Soundness:** 4
**Presentation:** 4
**Contribution:** 4
**Rating:** 10
**Confidence:** 3

**Summary:**

The paper considers a generalized Optimal Transport (OT) problem that has pairwise transport between multiple distributions with an added constraint to ensure the formation of a closed cycle. The authors present an iterative Sinkhorn algorithm to solve the Kantorovich formulation of the above-mentioned problem.
I think the paper is well-written and well-motivated. In particular, the alternative formulation of the TSP is very interesting, especially when one takes into account the performance and computation time. The problem itself is well-motivated with multiple applications. The numerical results often outperform, or at least stay competitive, with the state-of-the-art in all the examples.

I think there is a strong case for the acceptance of this paper.

**Strengths:**

- The quality of writing and presentation is high. Consequently, the results are presented in a clear and concise manner.
- While I do not think there is much originality/novelty (apart from the alternative TSP problem formulation) on the theoretical side, the improvements seen in the numerical simulations make a strong case for the significance of these results.

**Weaknesses:**

Not weaknesses per se, but I would like to see the following information included:
- I would like to see how sensitive the results are with respect to the optimization parameters, such as $\delta$.
- Table 1 should have computation time. Line 471 says that running time is presented in Table 1, but I do not see it.

**Questions:**

I don't have any questions in particular. However, I did notice a couple of typos (e.g., line 307 RCOT) and random capitalizations of words while reading, so I recommend that the authors perform thorough proofreading.

---

> ### Author Response · Authors · 2024-11-20
>
> Q1: How sensitive the results are with respect to the optimization parameters?
>
> A1: Thank you for highlighting the importance of analyzing the sensitivity of results to optimization parameters. We have conducted comprehensive sensitivity experiments for $\delta$ and $\epsilon$. The results, presented in as follows, demonstrate the robustness of our method.
>
> | $\delta$ | $\epsilon$ | ACC | CACC | CR |
> | ---- | ---- | ---- | ---- | ---- |
> | 0.001 | 1e-9 | 0.9412 | 0.8767 | 0.9158 |
> | 0.001 | 1e-10 | 0.9412 | 0.8767 | 0.9158 |
> | 0.01 | 1e-9 | 0.9442 | 0.8967 | 0.9475 |
> | 0.001 | 1e-11 | 0.9412 | 0.8767 | 0.9158 |
> | 0.01 | 1e-10 | 0.9442 | 0.8967 | 0.9475 |
> | 0.01 | 1e-11 | 0.9442 | 0.8967 | 0.9475 |
> | 0.1 | 1e-9 | 0.9382 | 0.9087 | 0.9951 |
> | 0.1 | 1e-10 | 0.9382 | 0.9087 | 0.9951 |
> | 0.1 | 1e-11 | 0.9382 | 0.9087 | 0.9951 |
>
> Q2: Some tyos and random capitalizations of words.
>
> A2: Thank you for your careful review. We have gone through the paper and corrected all such errors.

---

> > ### Comment · Reviewer_8SNv · 2024-11-24
> >
> > I thank the authors for clarifying some of my queries. I have no further comments.

---

### Author Response · Authors · 2024-11-27
**Any further questions?**

Thanks again for your comments! Is there any remaining concerns about our paper? We are more thandelighted to address any concerns/questions you may have.

---

### Author Response · Authors · 2024-12-03
**Summary of Initial Reviews and Responses**

We sincerely thank the reviewers for their time and valuable feedback. As the author-reviewer discussion wraps up, here's a summary of the reviews and our efforts during this phase:

| Reviewers' Concerns | Author Responses |
| --- | --- |
| How sensitive the results are with respect to the optimization parameters?    | We have conducted comprehensive sensitivity experiments for $\delta$ and $\epsilon$. The results are added to Table 6 in our paper, demonstrating the robustness of our method.                                                                                                                                                                                                                                                                                                                                                                                                                                      |
| Can the authors say anything about the analytic convengence of their methods? | Our algorithm is equivalent to the projection of the gradient descent algorithm. Please refer to the newly added reference in Section 4.2, where the convergence proof related to the our algorithm is discussed.                                                                                                                                                                                                                                                                                                                                                                                                    |
| Does a solution to the Monge formulation in Eq. (6) exist?   | A solution does exist. We can consider a feasible solution as follows: assume that $\{t_1, t_2, \dots, t_{K-1}\}$ are the solutions of the original MMOT problem. Then, given $x\in {\mathcal{X_1}}$ and $y=T_{K-1}T_{K-2}\cdots T_1(x)$, we can set $x=T_K(y)$. In this way, it satisfies the conditions and thus is a feasible solution. |
| Does the order of the probability measures $\alpha_k$ impact the results?     | It is obvious that the order of measures has no impact when $K < 3$. For $K=3, 4$, as shown in Section 4.3, the results before and after switching the order of measures are almost the same, demonstrating that the order of measures has little impact on the results for $K=3, 4$. It is important to note that in both the datasets we utilized and the majority of practical applications within our research domain, the value of $K$ typically does not exceed $4$. This practical constraint implies that the scenarios we are primarily concerned with are well-covered by our existing experimental setup. |
| At what rate is the Cycle-consistency is achieved?  | In theory, the cycle-consistency should be exactly satisfied. However, after applying relaxation techniques in our approach, it is approximately achieved. The Consistent Rate (CR) metric, which we have defined in Eq.(18) , provides insights into how closely our solutions approximate the ideal cycle-consistency.  |
| The setting of Hyper-parameter $\epsilon$. | We did not initially mention the setting of $\epsilon$ in the original paper. However, it can be tuned in a similar manner as $\delta'$. Specifically, one can replace $\lambda$ with $\epsilon$ in Algorithm 5 to perform the tuning process. We have now added the relevant description in the latest version of the paper to clarify this point. This way, readers can understand that the approach for adjusting $\epsilon$ is analogous to that of $\delta'$, providing a more comprehensive understanding of the hyperparameter settings in our methodology.      |
| The connection between MMOT and COT.     | The solution in MMOT is a tensor with the shape of $(n_1, n_2, \cdots, n_K)$, while COT's solution is a collection of couplings between pairs with the shape of $(n_1, n_2), (n_2, n_3), ...$. By the method newly added in Appendix H.2, MMOT can be transsformed to COT, which means that COT can leverage certain aspects of MMOT's framework while maintaining its own computational efficiency and applicability in the targeted problems.   |
| Are $(\mathbf{P_k})_{k=1}^K$ permutation matrices in Eq.(8)?                  | In our matching problem, the optimal $P_1,\dots,P_K$ are indeed permutation matrices. If the elements of $P_k$ were to be in the general range of $(0,1)$, the constraint $\prod P_k$ could not be satisfied. Only when $P_1,\dots,P_K$ are permutation matrices can this equality hold. The detailed proof has been added to Appendix A in our paper |

---

### Meta-Review · Area_Chair_xnHA · 2024-12-23

**Metareview:**

The authors consider the optimal transport problem and extend it for multiple input distributions. Instead of formulating as the popular multi-marginal OT (MMOT), the authors consider a cycle of OT over a pair of adjacent distributions. The Reviewers have mixed opinions on the submission. The considered problem may be an interesting contribution to the computational OT computational. However, the Reviewers raised several concerns on the order of the cycle (or whether one can get invariant property on such order), the motivation of the proposed approach (why COT instead of MMOT), the hardness of the cycle-consistency constraint (only satisfying for the Monge problem) which leads to the question how to relax it appropriately to the Kantorovich problem as in standard OT. The Reviewers also raised concerns on the 1D illustration which may mislead the intuition into high-dimensional case where permutation is required for the cycle-consistency. Thus, we think a major revision and further development are required for the proposed approach to plug it into the picture of computational OT. The authors can consider the comments of the Reviewers to improve the submission.

**Additional Comments On Reviewer Discussion:**

The Reviewers raised several significant concerns on the proposed approach for extending OT problem when there are many input measures as listed in the meta-review. Although the proposed approach may be interesting and a good additional piece for the computational OT picture, we think that a major revision and further development are required for the submission.

---

### Decision · Program_Chairs · 2025-01-22

Reject